# LinReTraCe: The Linear Response Transport Centre

Matthias Pickem [1], Emanuele Maggio [1], Jan M. Tomczak [1]

**1** Institute of Solid State Physics, TU Wien, Austria

June 14, 2022

## Abstract

We describe the "Linear Response Transport Centre" (LinReTraCe), a package for the simulation of transport properties of solids. LinReTraCe captures quantum (in)coherence effects beyond semi-classical Boltzmann techniques, while incurring similar numerical costs. The enabling algorithmic innovation is a semi-analytical evaluation of Kubo formulae for resistivities and the coefficients of Hall, Seebeck and Nernst. We detail the program's architecture, its interface and usage with electronic-structure packages such as WIEN2k, VASP, and Wannier90, as well as versatile tight-binding settings.

## Contents

arXiv:2206.06097v1 [cond-mat.mtrl-sci] 13 Jun 2022

# 1 Introduction

Signatures of external electrical fields, magnetic fields, thermal gradients, or combinations thereof can be used to monitor induced charge and energy currents in materials. Analyzing these *transport properties*, the ability to conduct charge, heat, or entropy, not only provides fundamental insight, but also quantifies potential functionalities.

    A key to the interpretation of measurements and to predict or optimize such properties are

transport *simulations*. Here, we describe the Linear Response Transport Centre LɪɴReTraCe (github.com/linretrace/linretrace), a software package that facilitates the computation of a variety of transport observables. The unique feature of LɪɴReTraCe is the treatment of thermal and lifetime broadening on an equal footing [1–3], while still incurring numerical costs as low as semi-classical Boltzmann approaches in the relaxation-time approximation. We exploit that linearizing the dynamics of many-body renormalizations (self-energy) allows for a semi-analytical (instead of numerical) evaluation of leading contributions in Kubo's linear response theory [1, 2, 4]. LɪɴReTraCe's principle input are electronic excitation energies and associated quasi-particle weights and lifetimes, as well as optical transition matrix elements. Being agnostic to the input's origin, LɪɴReTraCe can be used in a variety of settings, including electronic structures from tight-binding or Wannier projections [5], density functional theory (DFT) [6], many-body perturbation theory [7,8], dynamical mean-field theory (DMFT) [9,10], or approaches beyond [11, 12]. Scattering amplitudes and many-body renormalizations can be phenomenological, extracted from electronic self-energies (obtained, e.g., from DMFT), or could incorporate results from electron-phonon codes [13–19]. In this release we include interfaces to the DFT codes WIEN2k [20, 21] and VASP [22], the band interpolation tool of BoLtzTraP2 [23, 24], maximally localized Wannier functions of Wannier90 [25], as well as tools for general tight-binding systems. We further provide templates for the implementation of interfaces to other codes. With an emphasis on numerical accuracy and scalability, LɪɴReTraCe will be of value also for high-throughput studies [26–31] and for tight-binding descriptions of very large unit-cells.

## 1.1    Methodological context

To highlight the merits of LɪɴReTraCe, let us group previous packages for electronic transport properties of solids into two categories: semi-classical Boltzmann and Kubo linear response codes. Owing to their numerical efficiency and ease of handling, Boltzmann codes [13, 14, 23, 24, 32] have become popular tools. Typically, they are used in conjunction with band-theory, i.e., they utilize well-defined excitation energies. The selection of carriers participating in conduction is then solely determined from the *thermal* broadening (activation) via the Fermi function. In the usually employed relaxation-time approximation (RTA), electron scattering then only results in amplitude-scaling prefactors for a given momentum $\mathbf{k}$ and state $n$, e.g., for the conductivity $\sigma \propto \sum_{\mathbf{k}n} \Gamma_{\mathbf{k}n}^{-1} \times \cdots$, cf. Eq. (38). Often, the scattering rate $\Gamma$ is moreover assumed to be equal for all states and momenta. Then, the lifetime $\tau = \frac{\hbar}{2\Gamma}$ scales the Boltzmann conductivity globally, while the Seebeck and Hall coefficient become independent of $\tau$. With this assumption, insight into electronic transport can be gained without explicit knowledge of scattering amplitudes and their physical origin. As a consequence, Boltzmann transport kernels are relatively simple and the algorithmic complexity of, e.g., BoLtzTraP [23, 24] and BoLtzWann [32] (and the difference between them), lies in how they achieve convergence in the sampling of the Brillouin zone.

There are however circumstances, when the approximations inherent to band theory and the semi-classical treatment of transport fail. Inadequacies of band-theory for strongly correlated materials are well-documented: Examples not only include Mott insulators, where correlation effects fully invalidate the band-picture [33]. Also the electronic structure of correlated metals [34–37] and correlated narrow-gap semiconductors [38,39] are severely altered and have to be accounted for with methodologies that include dynamical renormalizations (self-energies). However, even if such many-body corrections are captured on the level of

electronic structure theory, plugging them into a semi-classical transport methodology may still lead to severe pathologies: Irrespective of the size of the scattering rate, the resistivity, the Seebeck, and the Hall coefficient of a clean semiconductor diverge in the zero temperature limit within Boltzmann's relaxation time approximation. A diverging activation law for the $T \to 0$ resistivity is physically admissible—but it is never observed. A diverging Seebeck coefficient, $|S(T \to 0)| \to \infty$, instead, violates the third law of thermodynamics (there can be no entropy transport at $T = 0$ for non-degenerate ground-states).

A quantum mechanical description of transport using Kubo's linear response theory [40] overcomes these artefacts by correctly treating effects of finite lifetimes (incoherence) of charge carriers [1, 2, 41]. To allow, beyond the *thermal*, also for a *lifetime* broadening of excitations, Kubo formulas require an integration over energies. This evaluation may become expensive for large systems and is hazardous at low temperatures and when the scattering rate is small. With LinReTraCe we conquer this bottleneck by performing frequency integrations *analytically* instead of numerically. This step becomes possible after linearizing the dynamics of many-body renormalizations (the self-energy $\Sigma$), yielding the LinReTraCe input: the scattering rate $\Gamma$, the quasi-particle weight $Z$, and possible static offsets $\Re\Sigma$. This approximation is warranted as the self-energy typically varies slowly inside the narrow energy-window probed by transport (a few $k_B T$).[1] Following common practice of Kubo implementations [35–37, 42–49], LinReTraCe neglects particle-hole scattering, so-called vertex corrections.[2] The ensuing analytical transport functions are then not only numerically inexpensive and stable, they also reveal valuable microscopic information: In particular they show that the scattering is not a mere prefactor (scaling the amplitude of conduction), but a relevant energy scale that has a complex interplay with other energies of the system, e.g., the charge gap in a semiconductor [2]. In all, LinReTraCe combines the best of both (Boltzmann & Kubo) worlds: an efficient and stable evaluation of transport observables that treats thermal and lifetime broadening on an equal footing. For a detailed derivation of the theory, see Ref. [2, 3].

## 1.2 Transport coefficients

Physical observables of transport processes are described by charge and heat currents

$$j_e^\alpha = \mathcal{L}_{11}^{\alpha\beta} E^\beta - \frac{1}{T} \mathcal{L}_{12}^{\alpha\beta} \partial_\beta T \tag{1}$$

$$j_q^\alpha = -\mathcal{L}_{21}^{\alpha\beta} E^\beta - \frac{1}{T} \mathcal{L}_{22}^{\alpha\beta} \partial_\beta T, \tag{2}$$

generated by an external electric field $\mathbf{E}$ and a temperature gradient $\nabla T$ perturbing the system. This non-equilibrium state can be described within linear response theory. There,

---

[1] Unless there are relevant pole-like structures in the self-energy, as is the case in Mott insulators.

[2] Vertex corrections mediated, e.g., by the Coulomb interaction (leading to, e.g., excitons or $\pi$-tons [50]) or disorder scattering (effects of, e.g., weak localization [51]), that dress the excited particle-hole pairs, are actively researched [50, 52–56], but are often small in practice [55]. Formally, they vanish in the (DMFT) limit of infinite dimensions [9], at least in the one-orbital case [57].

the coupling constants (the Onsager coefficients) become

$$
\mathcal{L}_{ab}^{\alpha\beta} = \frac{\pi\hbar e^{(4-a-b)}}{V} \sum_{\substack{n,m \\ \mathbf{k},\sigma}} \mathcal{K}_{ab}(\mathbf{k},n,m) M^{\alpha\beta}(\mathbf{k},n,m) \tag{3}
$$

$$
\mathcal{L}_{ab}^{B,\alpha\beta\gamma} = \frac{4\pi^2\hbar e^{(5-a-b)}}{3V} \sum_{\substack{n,m \\ \mathbf{k},\sigma}} \mathcal{K}_{ab}^{B}(\mathbf{k},n,m) M^{B,\alpha\beta\gamma}(\mathbf{k},n,m) \tag{4}
$$

with $a,b \in \{1,2\}$ defining the coupling in the Cartesian $\alpha,\beta,\gamma \in \{x,y,z\}$ direction and the summations running over all possible band combinations $(n,m)$, momenta in the Brillouin zone $\mathbf{k} \in \mathrm{BZ}$ and spins $\sigma$. The coefficients $\mathcal{L}_{ab}^{B,\alpha\beta\gamma}$ are the linear correction terms in the presence of a magnetic flux $(\mathbf{B})$ in the $\gamma$-direction. Employing the Onsager-Casimir relation

$$
\mathcal{L}_{ab}^{\alpha\beta}(\mathbf{B}) = \mathcal{L}_{ba}^{\beta\alpha}(-\mathbf{B}) \tag{5}
$$

on the expansion

$$
\mathcal{L}_{ab}^{\alpha\beta}(\mathbf{B}) = \mathcal{L}_{ab}^{\alpha\beta} + \mathcal{L}_{ab}^{B,\alpha\beta\gamma} B^{\gamma} + \mathcal{O}(\mathbf{B}^2), \tag{6}
$$

which we treat up to linear order in $\mathbf{B}$, one finds the connections

$$
\mathcal{L}_{ab}^{\alpha\beta} = \mathcal{L}_{ba}^{\beta\alpha} \tag{7}
$$

$$
\mathcal{L}_{ab}^{B,\alpha\beta\gamma} = -\mathcal{L}_{ba}^{B,\beta\alpha\gamma} \tag{8}
$$

when the $\mathbf{B}$-field is the only source of time-reversal symmetry breaking.[3] The Onsager coefficients combine the transport kernel functions, describing the (in our case: free) particle-hole propagation of the excited system,

$$
\mathcal{K}_{ab}(\mathbf{k},n,m) = \int_{-\infty}^{\infty} d\omega\, \omega^{(a+b-2)} \left(-\frac{\partial f}{\partial \omega}\right) A_{\mathbf{k}n}(\omega) A_{\mathbf{k}m}(\omega) \tag{9}
$$

$$
\mathcal{K}_{ab}^{B}(\mathbf{k},n,m) = \int_{-\infty}^{\infty} d\omega\, \omega^{(a+b-2)} \left(-\frac{\partial f}{\partial \omega}\right) A_{\mathbf{k}n}^2(\omega) A_{\mathbf{k}m}(\omega) \tag{10}
$$

with the optical matrix elements $M^{(B),\alpha\beta(\gamma)}$, describing the coupling of the external perturbation to the system. From them, the electrical conductivity $(\sigma)$, electrical resistivity $(\rho)$, Peltier coefficient $(\Pi)$, Seebeck coefficient $(S)$, thermal conductivity $(\kappa)$, Hall conductivity $(\sigma^B)$, Hall coefficient $(R_H)$, Nernst coefficient $(\nu)$[4], Hall mobility $(\mu_H)$ and its analogue, the thermal mobility $(\mu_T)$ [62] can be calculated. Note that, here, we limit the transport kernels to symmetric contributions. Anti-symmetric terms leading to anomalous transport from a non-trivial topological state [59, 63–65] are neglected. For a detailed dimensionality analysis of all involved quantities, see Appendix C.

---

[3]In magnetized materials the Onsager relations take the form of $\mathcal{L}_{ba}^{\beta\alpha}(H,M) = \mathcal{L}_{ab}^{\alpha\beta}(-H,-M)$ instead ($H$: magnetic field strength; $M$: magnetization density), leading to the anomalous Hall effect and a linear magneto resistance [58–60].

[4]We use the historical convention for the sign of the Nernst coefficient [61].

$$\sigma_{\alpha\beta} = \mathcal{L}_{11}^{\alpha\beta} \tag{11}$$

$$\rho_{\alpha\beta} = \left(\mathcal{L}_{11}^{-1}\right)^{\alpha\beta} \tag{12}$$

$$\Pi_{\alpha\beta} = -\left(\mathcal{L}_{11}^{-1}\right)^{\alpha i} \mathcal{L}_{12}^{i\beta} \tag{13}$$

$$S_{\alpha\beta} = -\frac{1}{T}\left(\mathcal{L}_{11}^{-1}\right)^{\alpha i} \mathcal{L}_{12}^{i\beta} \tag{14}$$

$$\kappa_{\alpha\beta} = \frac{1}{T}\left[\mathcal{L}_{22}^{\alpha\beta} - \mathcal{L}_{12}^{\alpha i}\left(\mathcal{L}_{11}^{-1}\right)^{ij}\mathcal{L}_{12}^{j\beta}\right] \tag{15}$$

$$\sigma_{\alpha\beta\gamma}^{B} = \mathcal{L}_{11}^{B,\alpha\beta\gamma} \tag{16}$$

$$R_{H,\alpha\beta\gamma} = \left(\mathcal{L}_{11}^{-1}\right)^{\alpha i} \mathcal{L}_{11}^{B,ij\gamma}\left(\mathcal{L}_{11}^{-1}\right)^{j\beta} \tag{17}$$

$$\nu_{\alpha\beta\gamma} = -\frac{1}{T}\left(\mathcal{L}_{11}^{-1}\right)^{\alpha i}\left[\mathcal{L}_{11}^{B,ij\gamma}\mathcal{L}_{12}^{jk} - \mathcal{L}_{12}^{B,ij\gamma}\mathcal{L}_{11}^{jk}\right]\left(\mathcal{L}_{11}^{-1}\right)^{k\beta} \tag{18}$$

$$\mu_{H,\alpha\beta\gamma} = \left(\mathcal{L}_{11}^{-1}\right)^{\alpha i} \mathcal{L}_{11}^{B,i\beta\gamma} \tag{19}$$

$$\mu_{T,\alpha\beta\gamma} = \left(\mathcal{L}_{12}^{-1}\right)^{\alpha i} \mathcal{L}_{12}^{B,i\beta\gamma} \tag{20}$$

## 2 Transport equations and optical elements

### 2.1 Kubo kernel expressions

The equations implemented in LinReTraCe result from the analytic integration of Eqs. (9-10), assuming a Lorentzian spectral function $A(\omega)$

$$A_{\mathbf{k}n}(\omega) = \frac{Z_{\mathbf{k}n}}{\pi}\frac{\Gamma_{\mathbf{k}n}}{(\omega - a_{\mathbf{k}n})^2 + \Gamma_{\mathbf{k}n}^2}. \tag{21}$$

This form is motivated by a linearization of the self-energy (for a discussion, see Ref. [2])[5]

$$\Sigma_{\mathbf{k}n}(\omega) \approx \Re\Sigma_{\mathbf{k}n}(0) + (1 - Z_{\mathbf{k}n}^{-1})\omega - i\Gamma_{\mathbf{k}n}^0. \tag{22}$$

Eq. (21) describes a quasi-particle peak of weight $Z_{\mathbf{k}n}$ with renormalized scattering rates $\Gamma_{\mathbf{k}n} = Z_{\mathbf{k}n}\Gamma_{\mathbf{k}n}^0$ and renormalized energies $a_{\mathbf{k}n} = Z_{\mathbf{k}n}(\varepsilon_{\mathbf{k}n}^0 - \mu + \Re\Sigma_{\mathbf{k}n}(0))$. Then, as detailed in Ref. [2], the kernels Eqs. (9,-10) can be calculated analytically. For the intra-band kernels ($n \equiv m$) one finds, with polygamma functions $\psi_n(z)$ [66] evaluated at $z = \frac{1}{2} + \frac{\beta}{2\pi}(\Gamma + ia)$:

---

[5]Eq. (22) shows the expansion (to linear order) of the self-energy around the Fermi level ($\omega = 0$). Typically, an expansion around the finite quasi-particle energy $a_{\mathbf{k}n}$ is more accurate and can be provided by the user.

$$\mathcal{K}_{11}(\mathbf{k}, n) = \frac{Z^2 \beta}{4\pi^3 \Gamma} \left[ \Re\psi_1(z) - \frac{\beta\Gamma}{2\pi} \Re\psi_2(z) \right] \tag{23}$$

$$\mathcal{K}_{12}(\mathbf{k}, n) = \frac{Z^2 \beta}{4\pi^3 \Gamma} \left[ a\Re\psi_1(z) - \frac{a\beta\Gamma}{2\pi} \Re\psi_2(z) - \frac{\beta\Gamma^2}{2\pi} \Im\psi_2(z) \right] \tag{24}$$

$$\mathcal{K}_{22}(\mathbf{k}, n) = \frac{Z^2 \beta}{4\pi^3 \Gamma} \left[ (a^2 + \Gamma^2)\Re\psi_1(z) + \frac{\beta\Gamma}{2\pi} \left( \Gamma^2 - a^2 \right) \Re\psi_2(z) - \frac{a\beta\Gamma^2}{\pi} \Im\psi_2(z) \right] \tag{25}$$

$$\mathcal{K}_{11}^B(\mathbf{k}, n) = \frac{Z^3 \beta}{16\pi^4 \Gamma^2} \left[ 3\Re\psi_1(z) - \frac{3\beta\Gamma}{2\pi} \Re\psi_2(z) + \frac{\beta^2\Gamma^2}{4\pi^2} \Re\psi_3(z) \right] \tag{26}$$

$$\mathcal{K}_{12}^B(\mathbf{k}, n) = \frac{Z^3 \beta}{16\pi^4 \Gamma^2} \left[ 3a\Re\psi_1(z) - \frac{3a\beta\Gamma}{2\pi} \Re\psi_2(z) - \frac{\beta\Gamma^2}{2\pi} \Im\psi_2(z) + \frac{a\beta^2\Gamma^2}{4\pi^2} \Re\psi_3(z) + \frac{\beta^2\Gamma^3}{4\pi^2} \Im\psi_3(z) \right] \tag{27}$$

$$\begin{aligned} \mathcal{K}_{22}^B(\mathbf{k}, n) = \frac{Z^3 \beta}{16\pi^4 \Gamma^2} \bigg[ &(3a^2 + \Gamma^2)\Re\psi_1(z) - \frac{\beta\Gamma}{2\pi}(3a^2 + \Gamma^2)\Re\psi_2(z) - \frac{a\beta\Gamma^2}{\pi}\Im\psi_2(z) \\ &- \frac{\beta^2\Gamma^2}{4\pi^2}(\Gamma^2 - a^2)\Re\psi_3(z) + \frac{a\beta^2\Gamma^3}{2\pi^2}\Im\psi_3(z) \bigg] \end{aligned} \tag{28}$$

The inter-band ($n \neq m$) kernels become

$$\begin{aligned} \mathcal{K}_{11}(\mathbf{k}, n, m) = &\frac{Z_1 Z_2 \beta}{2\pi^3 \left[ (a_1 - a_2)^2 + (\Gamma_1 - \Gamma_2)^2 \right] \left[ (a_1 - a_2)^2 + (\Gamma_1 + \Gamma_2)^2 \right]} \\ &\times \bigg[ \Re\Big\{ \left[ (a_1 - a_2)^2 + \Gamma_2^2 - \Gamma_1^2 - 2i\Gamma_1(a_2 - a_1) \right] \Gamma_2\psi_1(z_1) \Big\} \\ &+ \Re\Big\{ \left[ (a_2 - a_1)^2 + \Gamma_1^2 - \Gamma_2^2 - 2i\Gamma_2(a_1 - a_2) \right] \Gamma_1\psi_1(z_2) \Big\} \bigg] \end{aligned} \tag{29}$$

$$\begin{aligned} \mathcal{K}_{12}(\mathbf{k}, n, m) = &\frac{Z_1 Z_2 \beta}{2\pi^3 \left[ (a_1 - a_2)^2 + (\Gamma_1 - \Gamma_2)^2 \right] \left[ (a_1 - a_2)^2 + (\Gamma_1 + \Gamma_2)^2 \right]} \\ &\times \bigg[ \Re\Big\{ (a_1 - i\Gamma_1) \left[ (a_1 - a_2)^2 + \Gamma_2^2 - \Gamma_1^2 - 2i(a_2 - a_1)\Gamma_1 \right] \Gamma_2\psi_1(z_1) \Big\} \\ &+ \Re\Big\{ (a_2 - i\Gamma_2) \left[ (a_2 - a_1)^2 + \Gamma_1^2 - \Gamma_2^2 - 2i(a_1 - a_2)\Gamma_2 \right] \Gamma_1\psi_1(z_2) \Big\} \bigg] \end{aligned} \tag{30}$$

$$\begin{aligned} \mathcal{K}_{22}(\mathbf{k}, n, m) = &\frac{Z_1 Z_2 \beta}{2\pi^3 \left[ (a_1 - a_2)^2 + (\Gamma_1 - \Gamma_2)^2 \right] \left[ (a_1 - a_2)^2 + (\Gamma_1 + \Gamma_2)^2 \right]} \\ &\times \bigg[ \Re\Big\{ (a_1 - i\Gamma_1)^2 \left[ (a_1 - a_2)^2 + \Gamma_2^2 - \Gamma_1^2 - 2i(a_2 - a_1)\Gamma_1 \right] \Gamma_2\psi_1(z_1) \Big\} \\ &+ \Re\Big\{ (a_2 - i\Gamma_2)^2 \left[ (a_2 - a_1)^2 + \Gamma_1^2 - \Gamma_2^2 - 2i(a_1 - a_2)\Gamma_2 \right] \Gamma_1\psi_1(z_2) \Big\} \bigg], \end{aligned} \tag{31}$$

where $\psi_1(z_{1/2})$ is evaluated at $z_{1/2} = \frac{1}{2} + \frac{\beta}{2\pi}(\Gamma_{1/2} + ia_{1/2})$. These kernels represent a generalization of Eqs. (23-25) and per Eq. (9) obey band-swapping symmetry $\mathcal{K}_{ab}(\mathbf{k}, n, m) \equiv \mathcal{K}_{ab}(\mathbf{k}, m, n)$. See Appendix D for the generic intra-band limit ($f_m \to f_n$, $f = Z, \Gamma, a$).

The magnetic inter-band $(n \neq m)$ kernels evaluate to

$$
\begin{aligned}
\mathcal{K}_{11}^B(\mathbf{k}, n, m) = {} & \frac{Z_1 Z_2^2 \Gamma_1 \Gamma_2^2 \beta}{2\pi^4} \\
& \times \Bigg[ \Re\Big\{ \frac{1}{\Gamma_1} \psi_1(z_1) \frac{1}{[a_1 - a_2 - i(\Gamma_1 + \Gamma_2)]^2 \, [a_1 - a_2 - i(\Gamma_1 - \Gamma_2)]^2} \Big\} \\
& - \Re\Big\{ \frac{\beta}{4\pi\Gamma_2^2} \psi_2(z_2) \frac{1}{[a_2 - a_1 - i(\Gamma_1 + \Gamma_2)] \, [a_2 - a_1 + i(\Gamma_1 - \Gamma_2)]} \Big\} \\
& + \Im\Big\{ \frac{1}{\Gamma_2^2} \psi_1(z_2) \frac{(a_2 - a_1 - i\Gamma_2)}{[a_2 - a_1 - i(\Gamma_1 + \Gamma_2)]^2 \, [a_2 - a_1 + i(\Gamma_1 - \Gamma_2)]^2} \Big\} \\
& + \Re\Big\{ \frac{1}{2\Gamma_2^3} \psi_1(z_2) \frac{1}{[a_2 - a_1 - i(\Gamma_1 + \Gamma_2)] \, [a_2 - a_1 + i(\Gamma_1 - \Gamma_2)]} \Big\} \Bigg]
\end{aligned}
\tag{32}
$$

$$
\begin{aligned}
\mathcal{K}_{12}^B(\mathbf{k}, n, m) = {} & \frac{Z_1 Z_2^2 \Gamma_1 \Gamma_2^2 \beta}{2\pi^4} \\
& \times \Bigg[ \Re\Big\{ \frac{1}{\Gamma_1} \psi_1(z_1) \frac{(a_1 - i\Gamma_1)}{[a_1 - a_2 - i(\Gamma_1 + \Gamma_2)]^2 \, [a_1 - a_2 - i(\Gamma_1 - \Gamma_2)]^2} \Big\} \\
& - \Im\Big\{ \frac{1}{2\Gamma_2^2} \psi_1(z_2) \frac{1}{[a_2 - a_1 - i(\Gamma_1 + \Gamma_2)] \, [a_2 - a_1 + i(\Gamma_1 - \Gamma_2)]} \Big\} \\
& - \Re\Big\{ \frac{\beta}{4\pi\Gamma_2^2} \psi_2(z_2) \frac{(a_2 - i\Gamma_2)}{[a_2 - a_1 - i(\Gamma_1 + \Gamma_2)] \, (a_2 - a_1 + i(\Gamma_1 - \Gamma_2))} \Big\} \\
& + \Im\Big\{ \frac{1}{\Gamma_2^2} \psi_1(z_2) \frac{(a_2 - i\Gamma_2)(a_2 - a_1 - i\Gamma_2)}{[a_2 - a_1 - i(\Gamma_1 + \Gamma_2)]^2 \, [a_2 - a_1 + i(\Gamma_1 - \Gamma_2)]^2} \Big\} \\
& + \Re\Big\{ \frac{1}{2\Gamma_2^3} \psi_1(z_2) \frac{(a_2 - i\Gamma_2)}{[a_2 - a_1 - i(\Gamma_1 + \Gamma_2)] \, [a_2 - a_1 + i(\Gamma_1 - \Gamma_2)]} \Big\} \Bigg]
\end{aligned}
\tag{33}
$$

$$
\begin{aligned}
\mathcal{K}_{22}^B(\mathbf{k}, n, m) = {} & \frac{Z_1 Z_2^2 \Gamma_1 \Gamma_2^2 \beta}{2\pi^4} \\
& \times \Bigg[ \Re\Big\{ \frac{1}{\Gamma_1} \psi_1(z_1) \frac{(a_1 - i\Gamma_1)^2}{[a_1 - a_2 - i(\Gamma_1 + \Gamma_2)]^2 \, [a_1 - a_2 - i(\Gamma_1 - \Gamma_2)]^2} \Big\} \\
& - \Im\Big\{ \frac{1}{\Gamma_2^2} \psi_1(z_2) \frac{(a_2 - i\Gamma_2)}{[a_2 - a_1 - i(\Gamma_1 + \Gamma_2)] \, [a_2 - a_1 + i(\Gamma_1 - \Gamma_2)]} \Big\} \\
& - \Re\Big\{ \frac{\beta}{4\pi\Gamma_2^2} \psi_2(z_2) \frac{(a_2 - i\Gamma_2)^2}{[a_2 - a_1 - i(\Gamma_1 + \Gamma_2)] \, (a_2 - a_1 + i(\Gamma_1 - \Gamma_2))} \Big\} \\
& + \Im\Big\{ \frac{1}{\Gamma_2^2} \psi_1(z_2) \frac{(a_2 - i\Gamma_2)^2(a_2 - a_1 - i\Gamma_2)}{[a_2 - a_1 - i(\Gamma_1 + \Gamma_2)]^2 \, [a_2 - a_1 + i(\Gamma_1 - \Gamma_2)]^2} \Big\} \\
& + \Re\Big\{ \frac{1}{2\Gamma_2^3} \psi_1(z_2) \frac{(a_2 - i\Gamma_2)^2}{[a_2 - a_1 - i(\Gamma_1 + \Gamma_2)] \, [a_2 - a_1 + i(\Gamma_1 - \Gamma_2)]} \Big\} \Bigg]
\end{aligned}
\tag{34}
$$

where $\psi_n(z_{1/2})$ is, again, evaluated at $z_{1/2} = \frac{1}{2} + \frac{\beta}{2\pi}(\Gamma_{1/2} + i a_{1/2})$.

## 2.2 Boltzmann approximation

The semiclassical Boltzmann transport expressions can be obtained by expanding the polygamma functions of our approach around $\Gamma = 0$

$$\psi_n \left( \frac{1}{2} + \frac{\beta}{2\pi}(\Gamma + ia) \right) = \psi_n \left( \frac{1}{2} + \frac{i\beta a}{2\pi} \right) + \frac{\beta\Gamma}{2\pi}\psi_{n+1} \left( \frac{1}{2} + \frac{i\beta a}{2\pi} \right) + \mathcal{O}\left( \Gamma^2 \right), \qquad (35)$$

and recognizing that

$$f_{\text{FD}}(a) = \frac{1}{2} - \frac{1}{\pi}\Im\psi \left( \frac{1}{2} + \frac{i\beta a}{2\pi} \right) \qquad (36)$$

$$-\partial_a f_{\text{FD}}(a) = \frac{\beta}{2\pi^2}\Re\psi_1 \left( \frac{1}{2} + \frac{i\beta a}{2\pi} \right). \qquad (37)$$

The Boltzmann intra-band expressions are then recovered from the expansions as the leading terms in $\Gamma$. Our quantum mechanical formalism thus contains the semi-classical description as the coherent (infinite lifetime) limit. To assure that the Boltzmann inter-band kernels reduce to the Boltzmann intra-band expressions in the limit of degenerate states (with the same lifetime), they have to include terms beyond the leading order. Instead, if the Boltzmann approximation only takes into account the leading terms, the limit $a_1 \to a_2; \Gamma_1 \to \Gamma_2$ will yield inconsistent results. To our knowledge this ensemble of inter-band Boltzmann expressions has not been derived previously:

$$\mathcal{K}_{11}^{\text{Boltzmann}}(\mathbf{k}, n) = -\frac{Z^2}{2\pi\Gamma}\partial_a f_{\text{FD}}(a) \qquad (38)$$

$$\mathcal{K}_{12}^{\text{Boltzmann}}(\mathbf{k}, n) = -\frac{aZ^2}{2\pi\Gamma}\partial_a f_{\text{FD}}(a) \qquad (39)$$

$$\mathcal{K}_{22}^{\text{Boltzmann}}(\mathbf{k}, n) = -\frac{\left( a^2 + \Gamma^2 \right) Z^2}{2\pi\Gamma}\partial_a f_{\text{FD}}(a) \qquad (40)$$

$$\mathcal{K}_{11}^{\text{Boltzmann},B}(\mathbf{k}, n) = -\frac{3Z^3}{8\pi^2\Gamma^2}\partial_a f_{\text{FD}}(a) \qquad (41)$$

$$\mathcal{K}_{12}^{\text{Boltzmann},B}(\mathbf{k}, n) = -\frac{3aZ^3}{8\pi^2\Gamma^2}\partial_a f_{\text{FD}}(a) \qquad (42)$$

$$\mathcal{K}_{22}^{\text{Boltzmann},B}(\mathbf{k}, n) = -\frac{\left( 3a^2 + \Gamma^2 \right) Z^3}{8\pi^2\Gamma^2}\partial_a f_{\text{FD}}(a) \qquad (43)$$

$$\mathcal{K}_{11}^{\text{Boltzmann}}(\mathbf{k}, n, m) = -\frac{Z_1 Z_2}{\pi \left[(a_1 - a_2)^2 + (\Gamma_1 - \Gamma_2)^2\right]\left[(a_1 - a_2)^2 + (\Gamma_1 + \Gamma_2)^2\right]}$$
$$\times \left[\left[(a_1 - a_2)^2 + \Gamma_2^2 - \Gamma_1^2\right]\Gamma_2 \partial_{a_1} f_{\text{FD}}(a_1) + \left[(a_2 - a_1)^2 + \Gamma_1^2 - \Gamma_2^2\right]\Gamma_1 \partial_{a_2} f_{\text{FD}}(a_2)\right] \tag{44}$$

$$\mathcal{K}_{12}^{\text{Boltzmann}}(\mathbf{k}, n, m) = -\frac{Z_1 Z_2}{\pi \left[(a_1 - a_2)^2 + (\Gamma_1 - \Gamma_2)^2\right]\left[(a_1 - a_2)^2 + (\Gamma_1 + \Gamma_2)^2\right]}$$
$$\times \left[\left\{\left[(a_1 - a_2)^2 + \Gamma_2^2 - \Gamma_1^2\right]a_1 - 2\Gamma_1^2(a_1 - a_2)\right\}\Gamma_2 \partial_{a_1} f_{\text{FD}}(a_1)\right.$$
$$\left. + \left\{\left[(a_2 - a_1)^2 + \Gamma_1^2 - \Gamma_2^2\right]a_2 - 2\Gamma_2^2(a_2 - a_1)\right\}\Gamma_1 \partial_{a_2} f_{\text{FD}}(a_2)\right] \tag{45}$$

$$\mathcal{K}_{22}^{\text{Boltzmann}}(\mathbf{k}, n, m) = -\frac{Z_1 Z_2}{\pi \left[(a_1 - a_2)^2 + (\Gamma_1 - \Gamma_2)^2\right]\left[(a_1 - a_2)^2 + (\Gamma_1 + \Gamma_2)^2\right]}$$
$$\times \left[\left\{\left[(a_1 - a_2)^2 + \Gamma_2^2 - \Gamma_1^2\right](a_1^2 - \Gamma_1^2) - 4\Gamma_1^2 a_1(a_1 - a_2)\right\}\Gamma_2 \partial_{a_1} f_{\text{FD}}(a_1)\right.$$
$$\left. + \left\{\left[(a_2 - a_1)^2 + \Gamma_1^2 - \Gamma_2^2\right](a_2^2 - \Gamma_2^2) - 4\Gamma_2^2 a_2(a_2 - a_1)\right\}\Gamma_1 \partial_{a_2} f_{\text{FD}}(a_2)\right] \tag{46}$$

$$\mathcal{K}_{11}^{\text{Boltzmann},B}(\mathbf{k}, n, m) = -\frac{Z_1 Z_2^2}{2\pi^2 \Gamma_2}$$
$$\times \left[\Re\left\{\frac{2\Gamma_2^3}{\left[a_1 - a_2 - i(\Gamma_1 + \Gamma_2)\right]^2 \left[a_1 - a_2 - i(\Gamma_1 - \Gamma_2)\right]^2}\right\}\partial_{a_1} f_{\text{FD}}(a_1)\right.$$
$$+\Im\left\{\frac{2\Gamma_1\Gamma_2(a_2 - a_1 - i\Gamma_2)}{\left[a_2 - a_1 - i(\Gamma_1 + \Gamma_2)\right]^2 \left[a_2 - a_1 + i(\Gamma_1 - \Gamma_2)\right]^2}\right\}\partial_{a_2} f_{\text{FD}}(a_2)$$
$$\left.+\Re\left\{\frac{\Gamma_1}{\left[a_2 - a_1 - i(\Gamma_1 + \Gamma_2)\right]\left[a_2 - a_1 + i(\Gamma_1 - \Gamma_2)\right]}\right\}\partial_{a_2} f_{\text{FD}}(a_2)\right] \tag{47}$$

$$\mathcal{K}_{12}^{\text{Boltzmann},B}(\mathbf{k}, n, m) = -\frac{Z_1 Z_2^2 \beta}{2\pi^2 \Gamma_2}$$
$$\times \left[\Re\left\{\frac{2\Gamma_2^3(a_1 - i\Gamma_1)}{\left[a_1 - a_2 - i(\Gamma_1 + \Gamma_2)\right]^2 \left[a_1 - a_2 - i(\Gamma_1 - \Gamma_2)\right]^2}\right\}\partial_{a_1} f_{\text{FD}}(a_1)\right.$$
$$-\Im\left\{\frac{\Gamma_1\Gamma_2}{\left[a_2 - a_1 - i(\Gamma_1 + \Gamma_2)\right]\left[a_2 - a_1 + i(\Gamma_1 - \Gamma_2)\right]}\right\}\partial_{a_2} f_{\text{FD}}(a_2)$$
$$+\Im\left\{\frac{2\Gamma_1\Gamma_2(a_2 - i\Gamma_2)(a_2 - a_1 - i\Gamma_2)}{\left[a_2 - a_1 - i(\Gamma_1 + \Gamma_2)\right]^2 \left[a_2 - a_1 + i(\Gamma_1 - \Gamma_2)\right]^2}\right\}\partial_{a_2} f_{\text{FD}}(a_2)$$
$$\left.+\Re\left\{\frac{\Gamma_1(a_2 - i\Gamma_2)}{\left[a_2 - a_1 - i(\Gamma_1 + \Gamma_2)\right]\left[a_2 - a_1 + i(\Gamma_1 - \Gamma_2)\right]}\right\}\partial_{a_2} f_{\text{FD}}(a_2)\right] \tag{48}$$

$$\mathcal{K}_{22}^{\text{Boltzmann},B}(\mathbf{k}, n, m) = -\frac{Z_1 Z_2^2}{2\pi^2 \Gamma_2}$$

$$\times \left[ \Re\left\{ \frac{2\Gamma_2^3 (a_1 - i\Gamma_1)^2}{[a_1 - a_2 - i(\Gamma_1 + \Gamma_2)]^2 [a_1 - a_2 - i(\Gamma_1 - \Gamma_2)]^2} \right\} \partial_{a_1} f_{\text{FD}}(a_1) \right.$$

$$-\Im\left\{ \frac{2\Gamma_1 \Gamma_2 (a_2 - i\Gamma_2)}{[a_2 - a_1 - i(\Gamma_1 + \Gamma_2)][a_2 - a_1 + i(\Gamma_1 - \Gamma_2)]} \right\} \partial_{a_2} f_{\text{FD}}(a_2) \qquad (49)$$

$$+\Im\left\{ \frac{2\Gamma_1 \Gamma_2 (a_2 - i\Gamma_2)^2 (a_2 - a_1 - i\Gamma_2)}{[a_2 - a_1 - i(\Gamma_1 + \Gamma_2)]^2 [a_2 - a_1 + i(\Gamma_1 - \Gamma_2)]^2} \right\} \partial_{a_2} f_{\text{FD}}(a_2)$$

$$\left. +\Re\left\{ \frac{\Gamma_1 (a_2 - i\Gamma_2)^2}{[a_2 - a_1 - i(\Gamma_1 + \Gamma_2)][a_2 - a_1 + i(\Gamma_1 - \Gamma_2)]} \right\} \partial_{a_2} f_{\text{FD}}(a_2) \right]$$

In all, these Boltzmann approximations stem from a Taylor expansion of the terms in square brackets in Eqs. (23-28). Notably, there, terms linear in $\Gamma$ cancel exactly in the expansion of the *intra* band kernels, thus higher order terms 'only' become important if $\beta\Gamma$ is significant in size. Consequently Boltzmann results are accurate (comparable to Kubo) at elevated temperatures despite failing to fulfill fundamental theorems of thermodynamics in the zero temperature limit.

## 2.3 Quasi-particle renormalizations

The kernel expressions in Sec. 2.1, based on the self-energy linearization Eq. (22), clearly exhibit a non-trivial dependency on the quasi-particle renormalization: $Z$-factors emerge, both, as part of the overall pre-factor (à la Boltzmann) as well as in the argument of the polygamma functions via a renormalization of the energy ($a$) and the scattering ($\Gamma$):

Consider a *metallic* system with bare dispersion $\varepsilon_0(\mathbf{k})$ and a bare scattering rate $\Gamma_0$ at temperatures where the Boltzmann approximation is accurate. If the same quasi-particle renormalizations is applied to all states, a partial cancellation of the pre-factors in Eqs. (38-43) occurs due to renormalization of the scattering rate $\Gamma = Z\Gamma_0$. The Boltzmann response kernels then verify

$$\mathcal{K}_{ij}^{(B)} \propto Z a^{i+j-2} \partial_a f_{\text{FD}}(a). \qquad (50)$$

The conductivity kernel ($i = j = 1$) that is even in the energy $a = Z\epsilon_0$ is to good approximation unaffected by $Z$ due to a compensation of two effects: $Z$ simultaneously decreases the weights of the selected states, and pushes more states into the (thermal) selection window through band-narrowing.[6] Odd kernels ($\mathcal{L}_{12}, \mathcal{L}_{12}^B$) on the other hand distinguish between electron and hole contributions via the sign of $a$. Through this differentiation the overall summation will be tilted to either direction depending on the asymmetry of the system. Energy renormalization in this context then can be thought of as an amplification of this asymmetry,

---

[6]This is exact for $0 < Z \le 1$, in the limit of infinite bandwidth and a flat density of states. In realistic scenarios, however, there can be a notable $Z$-dependence for strong renormalizations and narrow band-widths, elevated temperatures, or a strongly energy-dependent density of states.

increasing the non-interacting signal[7]

$$\mathcal{K}_{12}^{(B)}(Z) \propto \frac{1}{Z}\mathcal{K}_{12}^{(B)}(Z = 1) \tag{51}$$

providing a correlation mechanism to boost the Seebeck [67,68] and Nernst coefficient, realized, e.g., in heavy-fermion systems [34]. The above arguments in general do not hold for insulating systems where we find a more nuanced interplay of band gap, energies, chemical potential and quasi-particle renormalization. At the very least $Z < 1$ will result in a band gap reduction $\Delta = Z\Delta_0$, which affects thermal activation, and hence conduction, exponentially.

## 2.4  Matrix elements

### 2.4.1  Dipole optical elements

Given the Fermi velocities (matrix elements of the momentum operator)

$$v_{\mathbf{k}nn'}^{\alpha} = \frac{1}{m}\left\langle \mathbf{k}n'|\mathcal{P}_\alpha|\mathbf{k}n \right\rangle \tag{52}$$

with $\alpha$ indicating a Cartesian direction, $m$ the electron mass, and $\langle \mathbf{r}|\mathbf{k}n\rangle = \chi_{\mathbf{k}n}(\mathbf{r})$ a band-momentum basis, the amplitude of optical dipole ($\mathbf{q} = 0$) transitions is given by

$$M^{\alpha\beta}(\mathbf{k}, n, m) = v_{\mathbf{k}nm}^{\alpha*}v_{\mathbf{k}mn}^{\beta} \tag{53}$$

and can be calculated within band-theory; for `WIEN2K`'s implementation see Ref. [69].

### 2.4.2  Peierls approximation

In tight-binding or model settings, in which there is no access to wavefunctions, the above matrix elements cannot be calculated. Instead, one couples the electromagnetic vector potential directly to the lattice fermions using the Peierls substitution approach [70]. Following this (approximate) procedure, Fermi velocities are momentum-derivatives of the (one-particle) Hamiltonian. Performing the derivative in the band-basis, there are only intra-band velocities, $v_{\mathbf{k}nm}^{\alpha} \propto \delta_{nm}v_{\mathbf{k}n}^{\alpha}$, for which

$$v_{\mathbf{k}n}^{\alpha} = \frac{1}{\hbar}\partial_{\mathbf{k}_\alpha}\varepsilon_n(\mathbf{k}), \tag{54}$$

and

$$M^{\alpha\beta}(\mathbf{k}, n, n) = v_{\mathbf{k}n}^{\alpha}v_{\mathbf{k}n}^{\beta}. \tag{55}$$

Using the band-curvatures $c_k^{\alpha\beta} = 1/\hbar\,\partial_{k_\alpha}\partial_{k_\beta}\varepsilon(\mathbf{k})$, also the matrix elements for magnetic quantities can be derived. One finds [71]

$$M^{B,\alpha\beta\gamma}(\mathbf{k}, n, n) = \varepsilon_{\gamma ij}v_{\mathbf{k}n}^{\alpha}c_{\mathbf{k}n}^{\beta i}v_{\mathbf{k}n}^{j} \tag{56}$$

---

[7]By assuming a linearized density of states centered around the thermal selection window, $D(\varepsilon) = D_0 + \alpha\varepsilon$, it is apparent that only the linear (constant) term is responsible for finite values of kernels that are odd (even) in the energy $a$. Quasi-particle renormalizations drop out for the constant term and amplify the effect of the linear term, leading to the increase of $\frac{1}{Z}$ of Eq. (51).

with $\varepsilon_{ijk}$ the Levy-Civita symbol in three dimensions. These expressions for $M^{(B)}$ are common to LINRETRACE, BOLTZTRAP and other codes.

Clearly, however, taking momentum-derivatives does not commute with a general basis transformation $U$: $\partial_k U^\dagger(\mathbf{k})H(\mathbf{k})U(\mathbf{k}) \neq U^\dagger(\mathbf{k})(\partial_k H(\mathbf{k}))U(\mathbf{k})$. Therefore, transport properties using the Peierls approach will be basis-dependent [72]. One can show [42] that the Peierls approximation is best (closest to the true dipole element) the more localized the basis is. In the tight-binding and Wannier mode, LINRETRACE therefore performs the momentum-derivative in the local/Wannier basis $\chi_{\mathbf{R}l}(\mathbf{r}) = \langle \mathbf{r}|\mathbf{R}l\rangle$:

$$v^\alpha_{\mathbf{k}ll'} = \frac{1}{\hbar}\partial_{k_\alpha}H^{ll'}(\mathbf{k}) - i(\rho^\alpha_{l'} - \rho^\alpha_l)H^{ll'}(\mathbf{k}) \tag{57}$$

where $H(\mathbf{k})$ is the Fourier transform of $H(\mathbf{R})$. In this generalized Peierls approach [42], the second term arises for unit-cells with more than one atom, with intra-cell coordinates $\rho_l$ of the atom hosting orbital $l$. This extra-term in particular assures that an arbitrary extension of the unit-cell (conventional cell or equivalent supercells) gives the same result as calculations for the primitive unit-cell. Velocities evaluated in the local basis (orbitals indexed with $l$) are then rotated into the band-basis (band-index $n$). Because of the mentioned non-commutation of momentum-derivative and basis-transformation, the generalized Peierls approach may yield inter-band transitions à la Eq. (53) that are absent in Eq. (54). Generalizations of the band-curvatures to the Wannier basis have not yet been derived, i.e., magnetic transport functions in LINRETRACE rely on Eq. (56).

# 3 Implementation

*Programming languages*
— Fortran 95 and Python 3
*Required dependencies*
— HDF5 ($\geq$ 1.12.1)
— h5py, numpy, scipy, ase ($\geq$ 3.18.0), spglib ($\geq$ 1.9.5)
*Optional dependencies*
— MPI (Fortran 95), matplotlib, boltztrap2 ($\geq$ 20.7.1), cmake, pip, git

## 3.1 Installation

The program package can be obtained directly from the command line

```
$ git clone https://github.com/linretrace/linretrace.git
```

or by downloading the files from the github page with a browser. The pre- and post-processing is handled via Python3 where HDF5 data files are interfaced with `h5py` and computationally costly work is performed with `scipy` and `numpy`. `Ase` [73] and `Spglib` [74] are used to detect crystal symmetries and create irreducible momentum meshes, if they are not provided by the DFT code. In order to achieve maximal operability, `BoltzTraP2` [24], as well as `matplotlib` are recommended. `BoltzTraP2` is interfaced to interpolate bands, necessary to generate band derivatives and curvatures used in the magnetic optical elements, and `matplotlib` is used for graphical plotting. A quick way to obtain all (necessary and optional) dependencies is via the pip package manager

```
$ pip install matplotlib h5py numpy scipy ase spglib
$ pip install boltztrap2
```

After that, the LINRETRACE Python scripts can be executed directly from the source folder. Alternatively, in the LINRETRACE folder, execute

```
$ python3 setup.py install
```

to make them globally available.[8] The main program is written in MPI parallelized Fortran95 for which an HDF5 installation is required.[9] Its installation needs a configuration file `make_include` in the `linretrace` folder. Here the Fortran compiler (FC, FCDG), Fortran flags (FFLAGS), Fortran precompiler flags (FPPFLAGS) and library paths (HDF5) are to be provided. For a single core installation FC and FCDG are identical. Multi core installations require the MPI compiler in FC to be compatible with FCDG. The MPI installation is 'activated' via the FPPFLAGS variable `-DMPI`. An exemplary make configuration `make_include` for a multi-core intel setup looks like

```
FC        = mpiifort
FCDG      = ifort
FFLAGS    = -O3
FPPFLAGS = -DMPI
HDF5      = -I/opt/hdf5-1.12.1_icc/include
HDF5     += -L/opt/hdf5-1.12.1_icc/lib -lhdf5_fortran -lhdf5hl_fortran
```

while a single-core gfortran setup could be

```
FC        = gfortran
FCDG      = gfortran
FFLAGS    = -O3
HDF5      = -I/opt/hdf5-1.12.1_gcc/include
HDF5     += -L/opt/hdf5-1.12.1_gcc/lib -lhdf5_fortran -lhdf5hl_fortran
```

The LINRETRACE executable is compiled with

```
$ make
```

which creates `bin/linretrace`.

```
$ make install
```

can be used to copy the linretrace binary to the `bin` folder in the user's home directory. For some compilers (notably **gfortran**) an explicit link to the dynamic HDF5 library is necessary before executing the binary, made available with

```
$ export LD_LIBRARY_PATH=/opt/hdf5-1.12.1_gcc/lib:$LD_LIBRARY_PATH
```

e.g., in the `.bashrc` file. A more detailed step-by-step installation guide is provided in the repository's `documentation/userguide.pdf`, also available here.

---

[8]If root access is not available, use `python3 setup.py install --user` instead. This command has to be rerun after every version update.

[9]The LINRETRACE installation includes an HDF5 wrapper written by one of the authors, see `https://github.com/linretrace/hdf5_wrapper` for more details. There, you can also find an installation guide for the required HDF5 library. Ensure that the HDF5 library and LINRETRACE use the same compiler.

## 3.2   Flowchart

LINRETRACE calculations follow the flow chart of Fig. 1 and require two mandatory files: an energy file and a config file. The former is a direct result of one of the various interfaces and contains all the necessary energies $\varepsilon(\mathbf{k}, n)$, optical elements $M^{(B),\alpha\beta(\gamma)}$, and other auxiliary data. Depending on the data source only some of these optical elements can be generated, as, e.g., the magnetic field optical elements require band velocities and curvatures, not available in stand-alone WIEN2K calculations. The transport calculation itself is configured via a config text file for which lconfig provides a minimal starting point. More elaborate options, e.g., impurity states, need to be added by hand, see Sec. 3.5. In the config file one has access to simplistic scattering rate (and quasi-particle weight) dependencies: polynomials in temperature. More control over these dependencies can be gained via a scattering file where one has the option to specify all available data points individually, see Sec. 3.4. The results of the transport calculation are then saved in the HDF5 output file containing all the Onsager coefficients $\mathcal{L}^{\alpha\beta}_{ab}$, $\mathcal{L}^{B,\alpha\beta\gamma}_{ab}$ among other auxiliary information, including the configuration, structure information, etc. lprint provides easy access to these data containers as well as their combinations that form the physical transport quantities. In the next sections we go into more detail about each step, for a quick set of instructions see the cheat sheet in Appendix A.

## 3.3   Energy File

The center piece of the LINRETRACE input is the LRTC energy file. It contains the band energies $\varepsilon(\mathbf{k}, n)$, associated optical elements $M^{\alpha\beta}(\mathbf{k}, n, m)$, $M^{B,\alpha\gamma}(\mathbf{k}, n, m)$ and all relevant information on the unit cell and the momentum mesh in HDF5 format. While the band-diagonal elements (energies; intra-band $M^{\alpha\beta}(\mathbf{k}, n, n)$; intra-band $M^{B,\alpha\beta\gamma}(\mathbf{k}, n, n)$) are saved as fully continuous datasets, we explicitly separate the off-diagonal elements into a tree structure. Each momentum point then is represented by an HDF5 group whose datasets contain only its personal directional and band dependencies. This is done so that the (momentum parallelized) Fortran program is able to load in the elements in a more efficient manner. The overall tree structure of these files then looks as follows ('...' signals more datasets in these groups).

```
/.bands                            Group
/.bands/energyBandMax              Dataset
...
/.kmesh                            Group
/.kmesh/nkp                        Dataset
...
/.unitcell                         Group
/.unitcell/volume                  Dataset
...
/energies                          Dataset
/kPoint                            Group
/kPoint/0000000001                 Group
/kPoint/0000000001/moments         Dataset
/kPoint/0000000001/momentsBfield   Dataset
...
/momentsDiagonal                   Dataset
/momentsDiagonalBfield             Dataset
```

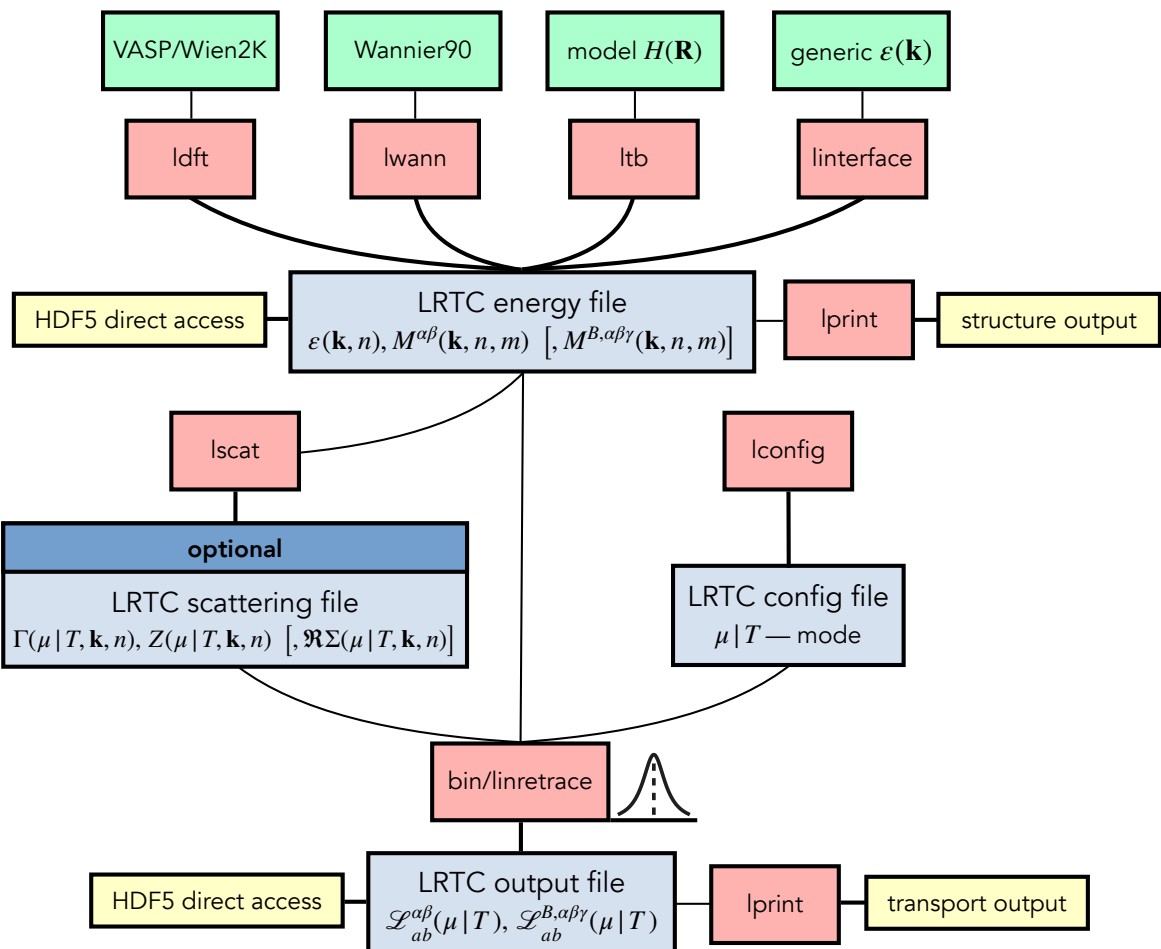

Figure 1: Flow chart of the LinReTraCe package: The energy file can be generated by interfacing various electronic structure codes or modelling your own dependencies. Combined with a config file and, *optionally*, full scattering dependencies, this constitutes the input of the core program `bin/linretrace`. The HDF5 output file can either be accessed effortlessly by `lprint` or via any external HDF5 library.

All the information that can be accessed in these pre-processed files can be listed with our multi-purpose interface `lprint`. Point the script at a specific input file and execute

```
$ lprint <LRTCinput file> list
```

We support the output of a structure information summary, the (spin-resolved) density of states and high-symmetry band paths via

```
$ lprint <LRTCinput file> info
$ lprint [-p] <LRTCinput file> dos
$ lprint [-p] <LRTCinput file> path
```

Where the `-p` (`--plot`) flag results in a graphical on-screen output. The path option relies on the `ase` library [73], which finds the high-symmetry points of the internal unit cell and displays them on-screen. Then, a custom **k**-path can be selected and plotted with

```
$ lprint [-p] <LRTCinput file> path GKMG # Gamma-K-M-Gamma
```

### 3.3.1 Density functional theory codes

Natively, we provide full support for the density functional theory codes WIEN2K [21] and VASP [22], and include connections to the BOLTZTRAP2 [24] library. In its basic form, to generate the LRTC energy file, the interface ldft simply has to be pointed at the folder of the (converged) DFT calculation

```
$ ldft <DFT folder> [--output lrtc-dft.hdf5]
```

This automatically detects if a valid calculation is present, initializes the according interface and detects the type of calculation that was performed (spin polarized, unpolarized, with or without spin-orbit-coupling).The --output flag sets the output file name, if none is provided we default to "lrtc-dft.hdf5".

**WIEN2k (version $\geq 18.x$).** We rely on the following files to be present[10]

- case.scf (number of electrons) [fallback: case.in2(c)][11]

- case.struct (atoms, symmetry operations)

- case.klist (k-points, multiplicities)

- case.energy (band structure)

The interface to the optical matrix elements (generated with x optic [69]) is done via the case.symmat files. The energy and optic file endings vary depending on the type of the calculation performed, which we detect automatically. Please note that ldft expects *exactly* 3, 6 or 9 columns in the symmat files. These correspond to the following case.inop configurations

- 3: Re xx, Re yy, Re zz

- 6: Re xx, Re yy, Re zz, Re xy, Re xz, Re yz,

- 9: Re xx, Re yy, Re zz, Re xy, Re xz, Re yz, Im xy, Im xz, Im yz

The read-in of the optical elements $M$ is activated via the --optic flag

```
$ ldft <WIEN2k folder> --optic
```

Alternatively, Peierls velocities can be used for $M^{\alpha\beta}$ via BoltzTraP2. The $M^{B,\alpha\beta\gamma}$ for magnetic quantities are only provided via the Peierls approach, see below.

---

[10]LINRETRACE assumes a serial WIEN2K run to be present. In case a parallelized calculation was performed, WIEN2K's join_vectorfiles tool must be used to combine the energy files.

[11]If the converged WIEN2K calculation was saved with save_lapw the case.scf file will not be present. In this case we fall back to case.in2 / case.in2c to retrieve the number of electrons. However, when using the BOLZTRAP2 interface, case.scf *must* be present, so you will have to copy the saved scf-file.

**VASP (version $\geq 5.x$).** Identically, simply point `ldft` to the `VASP` calculation folder where the file `vasprun.xml` must be present:

```
$ ldft <VASP folder>
```

If, as in `VASP`, no optical elements $M^{\alpha\beta}$ are provided by the electronic structure package, the interface to `BoltzTraP2` is required to generate the necessary input for transport calculations. For the magnetic elements, $M^{B,\alpha\beta\gamma}$, `BoltzTraP2` is always needed in conjunction with `ldft`.

**BoltzTraP2 interpolation.** We employ the Python3 library functionality of BoltzTraP2 to interpolate the band structure and construct derivatives and curvatures. These are then used to build the optical matrix elements according to Eq. (55-56). The interpolation is activated via

```
$ ldft <DFT folder> --interp [m]
```

The `--interp` optionally takes an additional interpolation parameter corresponding to the `BoltzTraP2 -m` option in the `btp2 interpolate` command. The default value is 3, meaning for every **k**-point of the input, `BoltzTraP2` tries to sample 3 additional irreducible **k**-points to generate the coefficients of the trigonometric polynomial representing the band structure. If the interpolation was done on an irreducible grid, we properly symmetrize the constructed optical elements $M^{\alpha\beta}$ and $M^{B,\alpha\beta\gamma}$, see Sec. 4.1. Since `VASP` does not provide the symmetry matrices at default settings, we extract the space group via the `ase` library [73]. For complicated structures and symmetry-broken calculations, we interactively ask the user to confirm the detected space group number or to assign the correct one.

**Band truncation.** For all input combinations we provide the possibility to truncate the energy levels around the Fermi level in order to reduce computational costs.

```
$ ldft <DFT folder> --trunc -5 +5
```

for example restricts the energy interval from $-5$ to $+5$eV around the DFT Fermi level and truncates every band that is fully outside of that interval. Please note that, contrary to, e.g., `WIEN2k`, we set the chemical potential of a fully gapped system to the center-point of the fundamental gap, i.e., $\mu_{\mathrm{DFT}} = [\min(E_{\mathrm{valence}}) + \max(E_{\mathrm{conduction}})]/2$.

### 3.3.2 Wannier90

Wannier90 [25], a program to calculate maximally localized Wannier functions [5], is based on minimizing the total spread of the Wannier function in real space. The output we are interested in are the real-space hopping parameters, $H_{ll'}(\mathbf{R})$, which we extract from the files

- `case.nnkp` (lattice vectors, **k**-points, projections)

- `case_hr.dat` (hopping $H_{ll'}(R)$)

- `case.wout` (units)

The multi-orbital Hamiltonian in reciprocal space and its derivatives (velocities $v$ and curvatures $c$) are then constructed from Fourier transforms

$$H_{ll'}(\mathbf{k}) = \sum_{\mathbf{R}} e^{i\mathbf{k}\cdot\mathbf{R}} H_{ll'}(\mathbf{R}) \tag{58}$$

$$v_{\mathbf{k}ll'}^{\alpha} = i \sum_{\mathbf{R}} R^{\alpha} e^{i\mathbf{k}\cdot\mathbf{R}} H_{ll'}(\mathbf{R}) \tag{59}$$

$$c_{\mathbf{k}ll'}^{\alpha\beta} = - \sum_{\mathbf{R}} R^{\alpha} R^{\beta} e^{i\mathbf{k}\cdot\mathbf{R}} H_{ll'}(\mathbf{R}) \tag{60}$$

where $R^{\alpha}$ is the Cartesian component $\alpha \in \{x, y, z\}$ of the unit-cell vector $\mathbf{R}$. For unit-cells with more than one atom, the velocities include an extra term from the generalized Peierls approach [42], see Eq. (57). We diagonalize the Hamiltonian matrices to go into the Kohn-Sham basis via

$$\varepsilon(\mathbf{k}) = U^{-1}(\mathbf{k}) H(\mathbf{k}) U(\mathbf{k}). \tag{61}$$

Applying the same unitary matrices onto the velocity and curvature matrices allows us to construct the transition matrices in the band basis. This procedure is more justified [42] than performing the momentum derivatives in the Kohn-Sham basis (see above). To interface a WANNIER90 calculation simply point `lwann` at the corresponding folder

```
$ lwann <Wannier90 folder> [--charge <N>] [--output lrtc-wann.hdf5]
```

where we try to extract the required charge `N` in the wannierized bands ourselves by rounding the calculated charge (at WANNIER90's $\mu = 0$) to its nearest integer value. This can be adjusted by providing the charge yourself with the optional `--charge` flag. The default output name `lrtc-wann.hdf5` can be adjusted with the `--output` flag. Since wannierizations are usually calculated on a restricted momentum grid, we provide options to refine the momentum grid directly:

```
$ lwann <Wannier90 folder> --kmesh <nkx> <nky> <nkz>
```

can be used to increase the reducible grid, where the new mesh has to conform to the old mesh's symmetry. Since reducible calculation can be costly we also provide an additional WIEN2K sub-interface that allows for setting up large *irreducible* meshes:

```
$ lwann <Wannier90 folder> --wien2k
```

Here the additional files

- `case.klist`

- `case.struct`

must be present in the same folder with the same file prefix (`case`) as used by WANNIER90, as is standard when using WIEN2WANNIER [75]. `Case.klist` (generated with WIEN2K via `x kgen`) provides a new (irreducible) momentum mesh with associated multiplicities while the symmetry operations in `case.struct` are used to symmetrize the calculated velocities and curvatures, see Sec. 4.1.

### 3.3.3 Tight binding models

In the same vein we provide an interface to generate input data from arbitrary tight-binding parameter sets. To this end we internally interface `spglib` [74] to generate the unit cell. Akin to WANNIER90 we require the primitive lattice vectors and the hopping parameters. In addition, a list of (in)equivalent atoms inside the unit cell is required to determine the unit-cell symmetries. With this information the tight binding input can be created via

```
$ ltb <tb file> <nkx> <nky> <nkz> <filling>
```

where we additionally provide the desired number of **k**-points in each direction and an initial band filling.[12] The underlying equations are identical to Sec. 3.3.2. An exemplary tight binding file looks as follows

```
begin hopping
#  a1 a2 a3    orb1 orb2  hopping.real [hopping.imag]
   0  0  0     1    1      0.3  # on site energy
  +1  0  0     1    1      1.0  # nearest neighbor hopping
   0 +1  0     1    1      1.0
  -1  0  0     1    1      1.0
   0 -1  0     1    1      1.0
end hopping

begin atoms
#  sort rx ry rz
   1    0  0  0     # fractional coordinates
end atoms

begin real_lattice
#  x   y   z
   5   0   0   # a1 lattice vector in units of Angstroem
   0   5   0   # a2
   0   0   1   # a3
end real_lattice
```

A number of different structures are saved as templates in the LINRETRACE repository. The hopping parameters are specified within the `begin hopping` and `end hopping` markers where each row represents the directional hopping amplitude along the lattice vectors (first three columns), the associated orbital indices (next two columns, numbering starts at 1) and finally the hopping amplitude (in units eV) in the last (two) column(s). The optional, 7th column allows for imaginary contributions to inter-orbital (orb1≠orb2) hoppings. Important to mention here is that, contrary to WANNIER90, we use the sign convention commonly used in the

---

[12]For insulators and semiconductors, the *nominal* filling should be given. The generated energy file can then be used also for doping studies, as the number of carriers (and also the band gap) can be modified via the `config.lrtc` file, see Appendix B.

strongly correlated electron community[13]

$$H(\mathbf{k}) = -\sum_{\mathbf{R}} e^{i\mathbf{k}\cdot\mathbf{R}}(1 - 2\delta_{\mathbf{R},\mathbf{0}}\delta_{l,l'})H_{ll'}(\mathbf{R}). \tag{62}$$

The fractional atomic positions in the unit cell are listed within the `begin atoms` and `end atoms` markers where each row represents one atom. The first columns counts the atomic 'species' starting at one, whereas the next three columns describe the fractional position within the unit cell with respect to the provided lattice vectors, i.e., all directional values are in the range $0 \leq r_i < 1$.[14] The orbital numbers in the hopping section are *not* connected to these atomic positions. Atoms are only relevant to determine the *symmetries* of the unit cell. Note that if the symmetry of the real-space hopping Hamiltonian is lower than the crystal-symmetry inferred from the atoms, the construction of the irreducible $\mathbf{k}$-mesh will fail. In that case "virtual" atoms have to be placed to manually break excess unit-cell symmetries. Finally, the lattice vectors in units of Ångstrom are saved as rows in between the `begin real_lattice` and `end real_lattice` markers.

Currently, the `ltb` interface only supports spin-unpolarized tight-binding files. A generalizations is straight forward. The filling argument is needed to pre-compute the chemical potential. Note that the counting is such that a fully filled orbital hosts 2 electrons. Again, for doping studies, the number of electrons can later be adjusted via the config file.

### 3.3.4   Generic interface

`linterface` contains the class `StructureFromArrays`, derived from the abstract base class `ElectronicStructure`. An object is instantiated with the number of $\mathbf{k}$-points in the three reciprocal lattice directions, list of real space lattice vectors, and the total charge in the system. After loading in the multiplicity, the energies, and optical elements in form of lists or `numpy` arrays with the corresponding `loadData` method, the output method `outputData` takes care of calculating the chemical potential, the setting of required flags and the file output itself. Since this interface is agnostic to the origin of the input, it can be used to interface any data source, including other density functional codes, dynamical mean field theory codes and so on. If developers write an interface to their electronic structure code, we will be happy to include it in the package.

## 3.4   Scattering File

For typical runs, information on the scattering rate and quasi-particle weights is set-up in the config file, see below. However, to include the full state- and momentum-dependence of the scattering amplitude $\Gamma$, the quasi-particle weight $Z$ and band-shifts $\Re\Sigma$, an otherwise optional LRTC scattering file has to be created. Therewith, $\Gamma$, $Z$, and $\Re\Sigma$ extracted from, e.g., a many-body calculation can be directly used to compute transport properties. To keep this route as generic as possible the user is required to interact with Python3 code, where at its core a scattering object is instantiated. First, the LRTC energy file from the previous section is read in to initialize states and momenta. Then, the only steps necessary are the definition of the calculation axis (chemical potential or temperature scan), the load in of user

---

[13]Hoppings are positive, with the gain of kinetic energy accounted for by a global minus sign, e.g., $\epsilon(\mathbf{k}) = -2t\sum_{\alpha=1,2,3}\cos(\mathbf{k}_\alpha)$ with $t > 0$.

[14]see `https://spglib.github.io/spglib/python-spglib.html` for details (last accessed 12.06.2022).

defined `numpy` arrays that describe the scattering dependencies, and the final output. The work flow is as follows:

1. Copy `lscat_template` from the installation folder into your working direction.

2. Insert the linretrace folder into the system path.[15]

3. Reference to correct energy file.

4. Define calculation axis ($\mu$ or $T$-scan).

5. Define scattering rates as a numpy array.
   Optionally: define quasi particle weights and/or band shifts.

6. Execute script to generate LRTC scattering file:

```
$ python3 lscat_template
```

A minimalistic script can look as simple as

---

[15]If the Python package was installed via `python3 setup.py install` this step is not necessary.

```
import sys
import numpy as np
sys.path.insert(0,'/home/user/linretrace') # Step 1: installation folder

from  scattering.fullscattering import FullScattering
scatobj = FullScattering('energies.hdf5')  # Step 2:
                                          # reference LRTC energy file

# main dependencies: spins, kpoints, bands
spins, nkp, nbands = scatobj.getDependencies()
kgrid              = scatobj.getMomentumGrid() # [ nkp, 3 ]
energies           = scatobj.getEnergies()     # [ spins, nkp, nbands ]
mudft              = scatobj.mudft              # chemical potential

# Step 3: define temperature range in Kelvin
nt = 100
scatobj.defineTemperatures(tmin = 10, \
                           tmax = 300, \
                           nt   = nt, \
                           tlog = True)

# Step 4: Create array that defines scattering rates in eV
gamma = np.zeros((nt, spins, nkp, nbands), dtype=np.float64) # Gamma
gamma[0,...]  = (energies-mudft)**2. / 10000.
gamma[1:,...] = gamma[0,...]

# Optional Step 5: Create arrays that define Z and ReSigma(0)
qpweight   = np.ones_like(gamma, dtype=np.float64)          # Z
bandshift  = np.zeros_like(gamma, dtype=np.float64)         # ReSigma

scatobj.defineScatteringRates(gamma, qpweight, bandshift)
scatobj.createOutput('scattering_file.hdf5')
```

How to handle the other dependencies is illustrated in code snippets in the provided `lscat` and `lscat_template`. In case scattering-rates and quasi-particle weights do not depend on band and momentum, and follow a simple temperature dependence, they can be more conveniently specified in the LinReTraCe configuration file, see the next section.

## 3.5 Configuring and running `LinReTraCe`

LinReTraCe is configured via a free format text configuration file. A minimalist starting point for this config file can be generated with `lconfig`. Here, through interactive questioning a basic config file is generated and saved as `config.lrtc`. From there, more elaborate options can be added manually. For more advanced settings, see Appendix B, for a full documentation of all possibilities, see the configuration specification in `documentation/configspec`.

In its basic form the configuration file sets the run mode and defines which quantities should be calculated and at which precision. LinReTraCe can either scan through a range

of temperatures for a given number of electrons (`RunMode = temp`) or scan through various chemical potentials (`RunMode = mu`) at fixed temperature. Further, one is also able to modify some information from the energy file on-the-fly, like the bandgap or the number of electrons in the system, e.g., for doping studies.

The run modes are then configured in their own respective section of the config file. Without a scattering file, the temperature mode is configured by specifying a temperature range and provides the option to include impurity states as well as homogeneous doping, see Sec. 4.2. Scattering rates can be specified for a simple polynomial temperature dependence ($\Gamma(T) = \Gamma_0 + \Gamma_1 T + \Gamma_2 T^2 + \cdots$) when acting on all bands and momenta equally. For the inclusion of arbitrary scattering rates, a scattering file needs to be created, see Section 3.4 and Appendix B.3. The chemical potential mode instead works at a fixed temperature and only requires information on the range over which the scan is performed and which scattering rate and quasi-particle renormalization to use.

For a temperature scan, the output of `lconfig` looks as follows

```
[General]                              # input/output configuration
RunMode         =   temp               # scan through temperatures
EnergyFile      =   lrtc-dft.hdf5      # any energy file from Sec. 3.3
OutputFile      =   output.hdf5        # output file name
BFieldMode      =   T                  # calculate L11B L12B L22B
Interband       =   F                  # T/F enable/disable inter-band
Intraband       =   T                  # enable intra-band
Boltzmann       =   T                  # Kubo AND Boltzmann responses
QuadResponse    =   T                  # kernel evaluation: quad precision
FermiOccupation =   F                  # digamma function as occupation

[TempMode]                             # temperature mode sub configuration
[[Scattering]]
TMinimum =  100.0                      # temperature range [K]
TMaximum =  700.0
TPoints  =  100
TLogarithmic = T                       # logarithmic temperature steps
ScatteringCoefficients = 1e-5 0 1e-8   # G0 G1 G2 ...
                                       # G(T) = G0 + G1 * T + G2 * T**2 ...
QuasiParticleCoefficients = 1          # renormalization Z0 ...
                                       # Z(T) = Z0 ...
```

The configuration of the chemical potential mode looks like

```
[General]
RunMode = mu
...
[MuMode]                                  # mu mode sub configuration
[[Scattering]]
Temperature   = 300.0                     # fixed temperature [K]
MuMinimum     = -5.0                      # chemical potential range [eV]
MuMaximum     = +5.0                      # with respect to mu_DFT
MuPoints      = 100
ScatteringRate       = 1e-5               # fixed Gamma [eV]
QuasiParticleWeight  = 1.0                # fixed Z [0 < Z <= 1]
```

Once the input and configuration has been prepared, LINRETRACE is run by executing the binary with the generated config file as argument. The MPI installation is invoked via

```
$ mpirun -np <cores> bin/linretrace config.lrtc
```

and the single core installation via

```
$ bin/linretrace config.lrtc
```

The internal program flow is listed in Table 1. During execution, first an options summary, then continuous run information is printed to the standard output. After a successful exit, the generated HDF5 output file contains the calculated Onsager coefficients as well as all relevant config information and some auxiliary datasets, see the next section.

### 3.6 Output File

In the HDF5 output file, all the calculated Onsager coefficients are saved as a combination of their identifier L11, L12, L22, L11B, L12B, L22B and the type of response that was used: `intra`, `inter`, `intraBoltzmann`, `interBoltzmann`. Non-magnetic datasets contain the momentum- and band-summed quantities with array shape [steps, spins, 3, 3] and magnetic datasets contain the momentum- and band-summed quantities with array shape [steps, spins, 3, 3, 3] where the last 2 (3) dimensions refer to the Cartesian directions $[\alpha, \beta]$ ($[\alpha, \beta, \gamma]$). Please note that we also include the possibility to output the Onsager coefficients with either their full dependency ($\mathbf{k}$, $n$), only momentum-$\mathbf{k}$-summed and only band-$n$-summed, see Appendix B. The standard tree structure of the output file looks as follows

|  | function name | file |
|---|---|---|
| initialize MPI interface | mpi_initialize | mpi_org.F90 |
| read config file | read_config | config.f90 |
| initialize HDF5 interface | hdf5_init | hdf5_wrapper.F90 |
| preprocess energy file | read_preproc_energy | input.f90 |
| *optional*: preprocess scattering file | read_preproc_scattering_hdf5 | input.f90 |
| or: | read_preproc_scattering_text | input.f90 |
| *optional*: read $\mu(T)$ | read_muT_hdf5 | input.f90 |
| or: | read_muT_text | input.f90 |
| distribute MPI **k**-load | mpi_genkstep | mpi_org.F90 |
| read $\varepsilon(\mathbf{k}, n)$, $M(\mathbf{k}, n, n)$ | read_energy | input.f90 |
| *optional*: read $M^B(\mathbf{k}, n, n)$ | read_energy | input.f90 |
| allocate data structures | allocate_response | response.F90 |
| *optional*: read $M(\mathbf{k}, n, m)$ | read_full_optical_elements | input.f90 |
| *optional*: read $M^B(\mathbf{k}, n, m)$ | read_full_magnetic_elements | input.f90 |
| initialize HDF5 output | output_auxiliary | output.F90 |
| **LOOP:**    **temperature** $T$ <br>          or **chemical potential** $\mu$ |  |  |
|   *optional*: read $\Gamma(k, n, \mu|T)$ | read_scattering_hdf5 | input.f90 |
|   *if necessary*: calculate $\mu(T)$ | find_mu | root.F90 |
|   calculate $\psi_n(\mathbf{k}, n)$ | calc_polygamma | response.F90 |
|   initialize data structures | initialize_response | response.F90 |
|   **LOOP: momentum k** |  |  |
|     calculate $\mathcal{K}_{ab} M^{\alpha\beta}$, $\mathcal{K}_{ab}^B M^{B,\alpha\beta\gamma}$ | response_intra_km | response.F90 |
|  | response_inter_km | response.F90 |
|  | response_intra_km_Q | response.F90 |
|  | $\vdots$ | response.F90 |
|     save / output $\mathcal{L}_{ab}^{\alpha\beta}$, $\mathcal{L}_{ab}^{B,\alpha\beta\gamma}$ | response_h5_output | response.F90 |
|  | response_h5_output_Q | response.F90 |
| output $\mu, T, n_e, n_h, \cdots$ <br> close MPI and HDF5 interface |  |  |

Table 1: Internal program flow of LinReTraCe.

```
/.config                    Group
/.quantities                Group
/.quantities/mu             Dataset
/.quantities/tempAxis       Dataset
...
/.scattering                Group
...
/.structure                 Group
/.structure/charge          Dataset
...
/.unitcell                  Group
/.unitcell/vol              Dataset
...
/L11                        Group
/L11/intra                  Group
/L11/intra/sum              Dataset
/L12                        Group
/L12/intra                  Group
/L12/intra/sum              Dataset
/L22                        Group
/L22/intra                  Group
/L22/intra/sum              Dataset
```

While extracting information directly from these files is possible (necessary for the full dependence output), it can be quite cumbersome. To this end we provide the post-processing interface `lprint`. Simply point `lprint` to the specific output file (or use "latest" to search for the latest valid output file) and execute

```
$ lprint <LRTCoutput file> list
```

to list all available physical datasets to print (`olist` to list all the Onsager coefficients). The configuration is listed via

```
$ lprint <LRTCoutput file> config
```

and datasets can be output (text/plot) by referring to the keys listed in the "list" option. To name a few selected datasets: the chemical potential can be plotted via

```
$ lprint [-p] <LRTCoutput file> mu
```

Transport tensors are evaluated from the Onsager coefficients via Eqs. 11-20, and can be extracted/plotted using the keywords from Tab. 2. `lprint` can be supplied with Cartesian directions chained after each other, e.g., the `xx` and `yy` entries of the conductivity tensor can be obtained via

```
$ lprint [-p] <LRTCoutput file> c-intra xx yy
```

If no direction is supplied, all combinations xx, xy, xz, yx, ... are plotted. Magnetic tensors, e.g., the Nernst coefficient require three directions, where the third defines the Cartesian direction of the applied magnetic field, cf. Eq. (6). E.g., for the Nernst coefficient:

| quantity | key | component |
|---|---|---|
| Onsager coefficients $\mathcal{L}_{ij}$ | Lij- | intra, inter |
| Onsager coefficients $\mathcal{L}_{ij}^B$ | LijB- | intra, inter |
| conductivity $\sigma$ | c- | intra, inter, total |
| resistivity $\rho$ | r- | intra, inter, total |
| Peltier $\Pi$ | p- | intra, inter, total |
| Seebeck $S$ | s- | intra, inter, total |
| power factor $S^2\sigma$ | pf- | intra, inter, total |
| thermal conductivity $\kappa$ | tc- | intra, inter, total |
| thermal resistivity $R_\lambda$ | tr- | intra, inter, total |
| Hall conductivity $\sigma^B$ | cb- | intra, inter, total |
| Hall coefficient $R_H$ | rh- | intra, inter, total |
| Nernst coefficient $\nu$ | n- | intra, inter, total |
| Hall mobility $\mu_H$ | muh- | intra, inter, total |
| thermal mobility $\mu_T$ | mut- | intra, inter, total |

Table 2: Syntax of `lprint`: The Onsager and transport tensors are accessed by combining the key and the component, e.g., `s-intra` for the intra-band Seebeck coefficient. The "total" contributions only exists if both intra- and inter-band transitions are present. For the Onsager coefficients Lij(B) possible indices are $ij \in \{11, 12, 22\}$.

```
$ lprint [-p] <LRTCoutput file> n-intra xyz
```

Spin resolved output is obtained by adding 'u' or 'd' to the directions (`uxx dxx` for the up and down component of the `xx` entry, respectively).
Identical datasets of different files can be compared with the `-c` (`--compare`) flag:

```
$ lprint [-p] <out1> L12-inter dzz -c <out2> <out3>
```

This flag in particular helps assessing the convergence of transport observables with the number of **k**-points, cf. Section 5.1.
Switching to the alternative print/plot axis, inverse temperature instead of temperature: $T[\text{K}] \to \beta[\text{eV}^{-1}]$, or, carrier concentration instead of chemical potential: $\mu[\text{eV}] \to n[\text{cm}^{-3}]$, is done via `-x` (`--axis`).

```
$ lprint [-p] -x <LRTCoutput file> s-total xx
```

The full functionality is described in the help option

```
$ lprint --help
```

# 4 Technical details

## 4.1 Matrix elements on irreducible grids

Periodic unit cells can be assigned a point group that is commonly represented by a set of $n$ square matrices $P_i$ ($\det P_i = \pm 1$) that describe all applicable symmetry operations acting

in real space. In reciprocal space this can be exploited as one can reduce the number of momenta necessary to represent the full Brillouin zone (for a symmetry reduction algorithm scaling linearly with the number of points see [76]). Each so-called irreducible **k**-point $\mathbf{k}_{\mathrm{irr}}$ then represents a set of momenta $\{\mathbf{k}_i\}$ that are all connected to each other via the transposed matrices $P_i^T$

$$\mathbf{k}_i = P_i^T \mathbf{k}_{\mathrm{irr}}. \tag{63}$$

The number of unique momenta generated from $\mathbf{k}_{\mathrm{irr}}$ is called its multiplicity $m$, where each point in the set will be generated exactly $\frac{n}{m}$ times. Naturally, the energies within this set remain unchanged

$$\forall i, \mathbf{k}_{irr}: \ \varepsilon(\mathbf{k}_{irr}) = \varepsilon(P_i^T \mathbf{k}_{irr}) \tag{64}$$

which then also applies to the kernel functions Eqs. (9-10). Crucially, owing to their directionality, this does not hold for the associated optical elements, band velocities and band curvatures, hence Eqs. (3-4) must not be evaluated directly on the irreducible grid. Instead, matrix elements have to be symmetrized: By averaging over all connected optical elements

$$M_{opt}^{\mathrm{symmetrized}}(\mathbf{k}_{\mathrm{irr}}) = \frac{1}{n}\sum_{i=1}^{n} M_{opt}(P_i^T \mathbf{k}_{\mathrm{irr}}) \tag{65}$$

one is able to absorb all the required symmetry information. Please note that these schemes require the momentum mesh to respect the same point group symmetries as the unit cell itself, e.g., a cubic crystal structure requires $n_{kx} = n_{ky} = n_{kz}$. If this were not the case, $P_i^T \mathbf{k}_{\mathrm{irr}}$ generates points outside the initial grid.[16]

While density functional theory codes like `WIEN2K` provide dipole matrix elements on an irreducible grid, optical elements as listed in Sec. 2.4.2 need to be symmetrized explicitly. Since Eq. 65 relies on information from the full Brillouin, the symmetrization is implemented via real-space rotations. Band velocities then transform as

$$v(P_i^T \mathbf{k}_{\mathrm{irr}}) = \left[K^{-1} P_i^{-1} K\right] v(\mathbf{k}_{\mathrm{irr}}) \tag{66}$$

whereas band curvatures $c$ transform as

$$c(P_i^T \mathbf{k}_{\mathrm{irr}}) = \left[K^{-1} P_i^{-1} K\right] c(\mathbf{k}_{\mathrm{irr}}) \left[K^{-1} P_i^{-1} K\right]^T \tag{67}$$

as they correspond to a single and twofold momentum derivative, respectively. In the same vein the optical elements $M$ themselves transform as the curvatures in Eq. (67). Here $K$ is the matrix formed by the reciprocal lattice vectors (the rows of $K$). This transforms the point group matrix into Cartesian directions and is explicitly necessary for non-orthogonal unit cells. The generated velocities and curvatures are then combined to optical elements via Eqs. (55-56), over which the symmetrization is performed. To showcase the correctness of these equations, see Fig. 2 where we compare a graphene-inspired honeycomb lattice on a reducible and corresponding irreducible momentum grid.

---

[16]An incommensurate reducible grid could still be reduced, by deselecting invalid momenta. Also an exact symmetry mapping of every single irreducible **k**-point to all $m$ connected momenta would solve this issue. Our current implementation, however, requires the user to make a sensible choice for the grid.

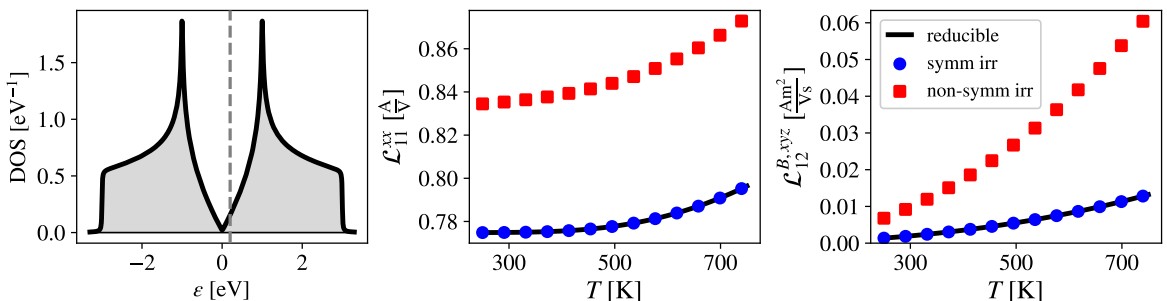

Figure 2: Reducible calculations compared to calculations on an irreducible grid with and without optical element symmetrization. Left: Density of states of the honeycomb lattice with hopping of $t_{AB} = t_{BA} = 1$eV and lattice length $|a| = 1$Å. Middle and right: Onsager coefficients $\mathcal{L}_{11}^{xx}$ and $\mathcal{L}_{12}^{xyz}$ performed for $180 \times 180$ reducible (8191 irreducible) **k**-points at a constant chemical potential of $\mu = 0.20$eV (vertical dashed line) and scattering rate $\Gamma = 10^{-5}$eV. Only a properly symmetrized irreducible grid leads to consistent data. Note that for this 2D system, Onsager coefficients needed to be multiplied with the (fictitious) $c$-lattice constant to yield proper units, as, e.g., $\mathcal{L}_{11}$ is linked to a conductance $[\sigma^{2D}] = 1/\Omega$ instead of a conductivity $[\sigma] = 1/(\Omega m)$, see Appendix C.3.

## 4.2 Chemical potential search

Determining the chemical potential is a common root finding problem: The numerical search of $\mu$ can be represented by

$$N - \sum_{\mathbf{k},n} f(\varepsilon_{\mathbf{k},n} - \mu) \stackrel{!}{=} 0. \tag{68}$$

For the case of no band renormalizations ($Z \equiv 1$) the occupation $f(\varepsilon_{\mathbf{k},n} - \mu)$ is either determined from the Fermi function

$$f(\varepsilon_{\mathbf{k},n} - \mu) = f_{\mathrm{FD}}(\varepsilon_{\mathbf{k},n} - \mu) = \frac{1}{1 + e^{\beta(\varepsilon_{\mathbf{k},n} - \mu)}} \tag{69}$$

or from the lifetime-broadened spectrum Eq. (21), entailing (see Ref. [2, 4])

$$f(\varepsilon_{\mathbf{k},n} - \mu) = \frac{1}{\pi} - \Im\psi\left(\frac{1}{2} + \frac{\beta}{2\pi}(\Gamma_{\mathbf{k},n} + i(\varepsilon_{\mathbf{k},n} - \mu))\right). \tag{70}$$

In some cases, employing root-finding algorithms on Eq. (68) can lead to severe problems. While metallic systems suffer mostly from too coarse momentum grids, gapped systems tend to exhibit massive numerical instabilities.[17] Due to the additional $\Gamma$-smearing in our formalism, this problem is absent for reasonably large scattering rates ($\Gamma \geq 10^{-6}$eV) and reasonable band gaps ($\Delta < 10$eV) at all temperatures when using Eq. (70). Using the Fermi-Dirac distribution for insulators, on the other hand, the root-finding is strongly restricted in the temperatures that can be safely captured, irrespective of the numerical accuracy, see double and quadruple precision calculations (blue and green lines) in Fig. 3, respectively. In order to circumvent

---

[17]Which is why Boltzmann codes typically use a fixed chemical potential instead of searching for it.

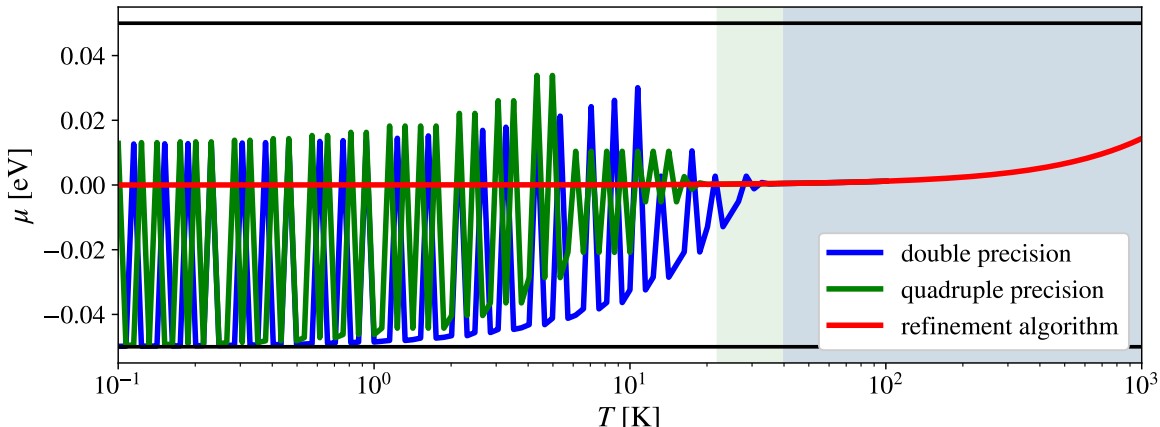

Figure 3: Chemical potentials in a gapped system determined via Eq. (68) with double (blue) and quadruple precision (green) compared to the chemical potential via the reformulated Eq. (71) (red). In all three cases the Fermi distribution is used. For the given band gap $\Delta = 0.1$eV the 'standard' root finding breaks down below $T = 40$K and $T = 10$K as the floating point accuracy is exhausted at double and quadruple precision, respectively. The reformulated problem on the other hand is stable down to $T = 0.05$K (below plotted range), see text.

this problem the root-finding problem can be reformulated to

$$\underbrace{\sum_{\mathbf{k},n\geq\text{CB}} f(\varepsilon_{\text{k},n} - \mu)}_{\text{activated electrons}} - \underbrace{\sum_{\mathbf{k},n\leq\text{VB}} f(-(\varepsilon_{\text{k},n} - \mu))}_{\text{activated holes}} \overset{!}{=} 0 \qquad (71)$$

in fully gapped systems with $\mu$ inside the gap: The chemical potential is determined by balancing the electrons in the conduction bands with the holes in valence bands. As a consequence one is not limited by machine precision anymore and can exploit the full floating point range. Nonetheless, due to finite bit length, temperatures are still bounded. The lowest possible achievable temperature corresponds to resolving density contributions down to the smallest positive number representable in quadruple precision: $2^{-16494}$. If the occupation is determined via the Fermi function and the chemical potential is in the middle of the band gap $\Delta$, it follows from

$$\frac{1}{e^{\beta\varepsilon} + 1} \approx e^{-\beta\varepsilon} = e^{-\beta\frac{\Delta}{2}}$$

that the lowest temperature bound is

$$T^{\mu}_{\text{bound}}[\text{K}] = \frac{\Delta[\text{eV}]}{2\ln(2)16494 k_B} \approx 0.5\Delta[\text{eV}]. \qquad (72)$$

The chemical potential determined via this refined root-finding problem is also illustrated in Fig. 3.

### 4.2.1 Impurity levels

LinReTraCe allows the inclusion of passive impurity states: Neglecting explicit contributions to the transport functions, the charge of these extra states affects the transport data merely

through the position of the chemical potential. The above refinement algorithm is especially important when employing impurity states inside the gap.[18] Including impurity states is straight forward:

1. calculate (intrinsic) electron occupation $f(\mu)$ according to Eq. (69) or Eq. (70)

2. calculate (extrinsic) impurity contribution $n_{\text{imp}}$ according to Eq. (73) or Eq. (74)

3. calculate total occupation: $n(\mu) = f(\mu) - n_{\text{imp}}^D + n_{\text{imp}}^A$ (given a donor (acceptor) level, the chemical potential has to increase (decrease) to compensate)

4. determine $\mu$ according to Eq. (68) or Eq. (71)[19]

Here the impurity contribution differs between donor (D) and acceptor (A) levels

$$n_{\text{imp}}^D = \frac{\rho_D}{1 + g e^{\beta(\mu - E_{\text{imp}}^D)}} \tag{73}$$

$$n_{\text{imp}}^A = \frac{\rho_A}{1 + g e^{\beta(E_{\text{imp}}^A - \mu)}} \tag{74}$$

where $\rho^{D/A}$ is the impurity density, $g$ is the impurity degeneracy and $E_{\text{imp}}$ is the impurity position. For a donor level in the vicinity of the conduction band edge the chemical potential has to increase to compensate for the additional impurity occupation. Fig. 4 illustrates the effect for varying impurity densities. Using the Fermi distribution for intrinsic states, the chemical potential approaches the center point between the impurity level $E_{\text{imp}}^D$ and the closest conduction state (instead of the band-gap middle point, realized for $\rho_D = 0$). From a transport perspective the effective band gap transforms from $\Delta = E_c - E_v$ at high temperatures to $\Delta = E_c - E_{\text{imp}}$ at low temperatures. The transition temperature is controlled by the impurity density and partially by the degeneracy of the impurity level. In addition to single impurity levels LinReTraCe also offers finite-size impurity bands with various shapes including box (constant), half-circle, and squared sine, see Appendix B.2.

### 4.2.2   Homogeneous doping

Besides explicit impurity states, LinReTraCe also supports generic doping. Contrary to impurity levels where the chemical potential converges for $T \to 0$ to a point inside the gap, any global doping forces the chemical potential eventually to move outside the gap. Leading up to this, the root finding works identical as in Sec. 4.2.1. Here the 'impurity contribution' is simply the doping which, now, is not affected by temperature and the position of the chemical potential. Please note that, technically, this kind of doping is more nuanced than a simple change of the total electron occupation. For the refinement algorithm to work inside the gap, an underlying integer filling is mandatory. Thus instead of changing the filling, we introduced an explicit `Doping` keyword in the config, see Appendix B.

    In Fig. 5 we showcase the same underlying band-structure as in Sec. 4.2.1 with various electron doping levels. Note the differences to Fig. 4: For the largest shown doping level $(\delta_e = 10^{-6})$ the crossing into the conduction band (shaded gray) already happens at around $T = 100K$.

---

[18]Without them, one could approximate the chemical potential obtained via the Fermi function, by extrapolating $\mu$ towards the band-gap mid-point at $T = 0$ [4].

[19]The reformulated root-finding problem of Eq. (71) becomes: $\sum_{\mathbf{k},n \in \text{CB}} f(\varepsilon_{\mathbf{k},n} - \mu) - \sum_{\mathbf{k},n \in \text{VB}} f(-(\varepsilon_{\mathbf{k},n} - \mu)) - n_{\text{imp}}^D + n_{\text{imp}}^A \overset{!}{=} 0$.

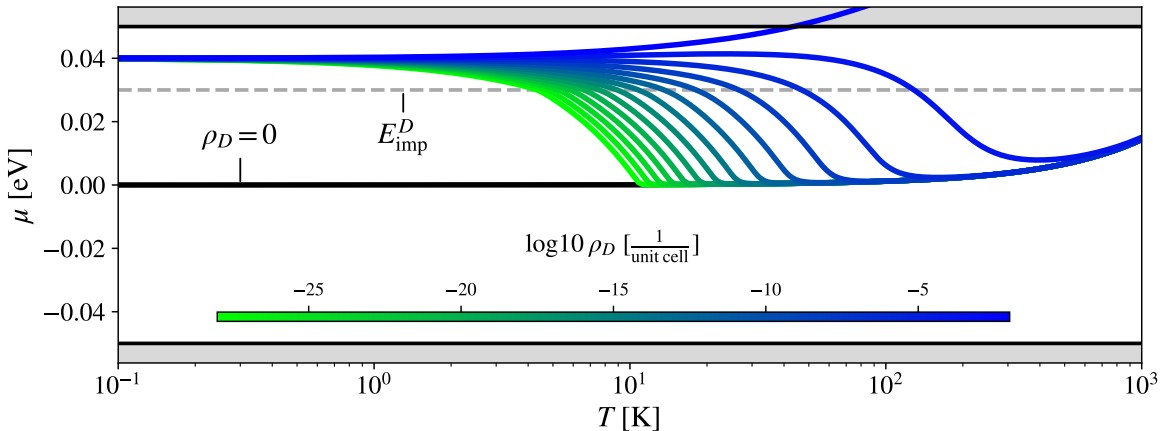

Figure 4: Comparison of the chemical potential for different impurity densities $\rho_D$ for a donor level located at $E^D_{\text{imp}}$ inside a semiconducting gap of 100meV. Intrinsic states were chosen to be described by the Fermi distribution ($\Gamma = 0$).

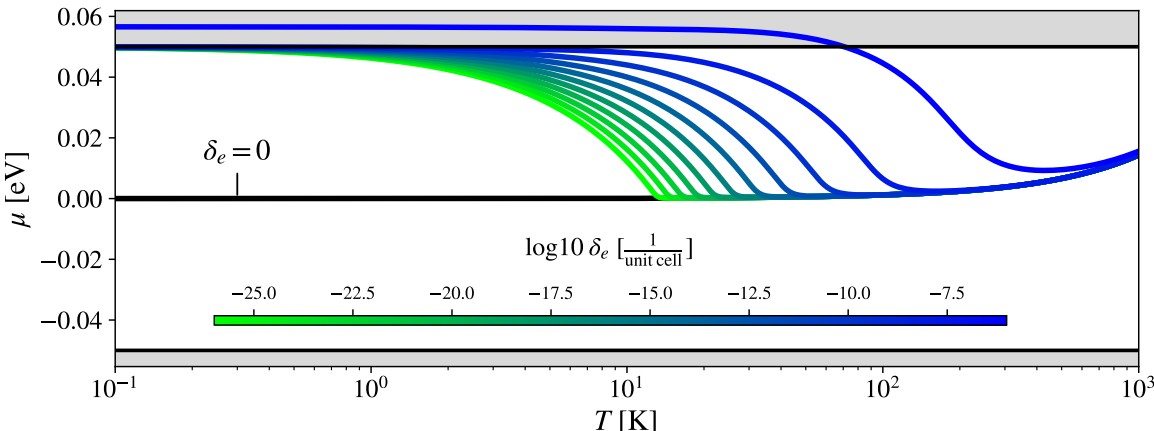

Figure 5: Comparison of the chemical potential for different levels of electron doping $\delta_e$ in a semiconductor. This example uses the Fermi distribution ($\Gamma = 0$) to find the chemical potential.

### 4.3 Polygamma evaluation

While all Fermi-function related equations can be implemented efficiently with native Fortran functions up to quadruple precision, the evaluation of the digamma and polygamma functions is more delicate. The digamma function is the derivative of the natural logarithm of the Gamma function whose series representation is closely related to the harmonic numbers

$$\psi(z) = \frac{d}{dz} \ln \Gamma(z) = \frac{\Gamma'(z)}{\Gamma(z)} \tag{75}$$

$$\psi(z) = -\gamma + \sum_{n=1}^{\infty} \left( \frac{1}{n} - \frac{1}{n+z-1} \right) \tag{76}$$

where $\gamma$ is the Euler–Mascheroni constant. The polygamma function $\psi_m$ $(m > 0)$ is the $m^{\text{th}}$ derivative of the digamma function whose series expansion follows directly from the differentiation

$$\psi_m(z) = \frac{d^m}{dz^m}\psi(z) \qquad m \in \mathbb{N}^+ \tag{77}$$

$$\psi_m(z) = (-1)^{m+1}m! \sum_{n=1}^{\infty} \frac{1}{(n+z-1)^{m+1}}. \tag{78}$$

Eqs. (76,78) are used extensively in our analytic derivation of the transport kernels [2]. Numerical implementations instead use different expressions depending on the location of the argument $z$ in the complex plane. In particular, for large $\Re(z)$, an asymptotic Bernoulli expansion makes the evaluation very efficient [66]:

$$\psi(z) = \ln(z) - (2z)^{-1} - \sum_{k=1}^{n-1} \frac{B_{2k}}{2k} z^{-2k} + \mathcal{O}(z^{-2n}) \qquad |\arg(z)| \leq \pi - \epsilon, \, \epsilon > 0 \tag{79}$$

with the Bernoulli numbers $B_{2k}$. In LinReTraCe, we adapt the cernlib [77] Fortran routine wpsipg (v1.2) for polygamma functions with complex argument, originally described by Kölbig [78]. We upgraded the routine to quadruple precision and commensurately increased the Bernoulli expansion order (to up to $k = 16$).[20]

## 4.4 Code scaling

As the code is meant to be a hybrid—designed to solve models *and* large realistic materials—the scaling behavior of the parallelization is important: LinReTraCe is parallelized over the number of Brillouin zone momenta $n_k$ (see Sec. 5.1 for a momentum convergence test). The resulting scaling is illustrated in Fig. 6. As expected, a purely linear behavior (runtime $t \propto n_k$) emerges.[21] While a single core installation experiences almost no overhead, the MPI implementation requires roughly 200 data points per core to become efficient. The quadruple precision evaluation of kernels roughly doubles the runtime.

In intra-band calculations the runtime will necessarily further scale linearly with the number of bands, while the inter-band portion of the code will scale with the square of the number of bands, as each band permutation must be evaluated. In normal circumstances, i.e., for small primitive unit cells, the number of bands is usually limited to between $\mathcal{O}(10)$ and $\mathcal{O}(100)$ where necessarily the momentum mesh must remain dense. On the contrary, in super cells one reallocates the computational load from the momentum mesh to the number of bands. The increased unit cell size requires fewer momenta to achieve convergence, which is counterbalanced by an increased number of atoms in the cell and thus an increased number of bands. To combat this intrinsic scaling problem one is encouraged to truncate the number of bands to a specific energy range around the Fermi level, see Section 3.3.1. Super cells especially benefit from this drastic decrease of the number of bands while maintaining numerical accuracy.

---

[20]At $T = 5.8$K and $\Gamma = 0$ this setting reproduces the analytical Fermi function $f_{FD}(\omega)$ with an error of less than $10^{-27}$ for any $\omega$.

[21]The calculations contain 100 temperature steps and were performed on a single node of the Vienna Scientific Cluster 4. Each node consists of two sockets with one Intel Xeon Platinum 8174 Processor @ 3.1 GHz (formerly Skylake) on each socket, leading to 48 physical cores. We used the Intel Fortran compiler ifort and mpiifort (2020 release) for the single core and multi core installation, respectively, with -O3 optimization.

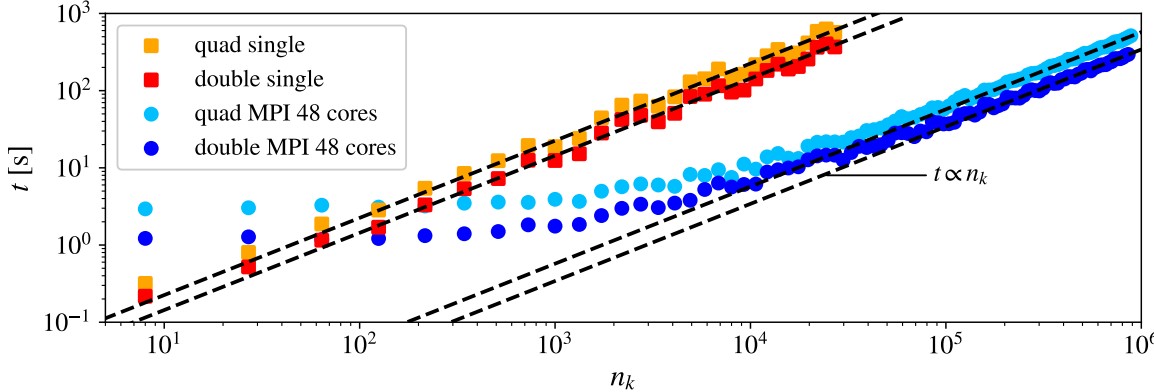

Figure 6: Runtime $t$ of a one-band model over a wide range of momentum grids with $n_k$ points. A clear linear scaling emerges which is somewhat delayed in the MPI runs due to the communication and input overhead. "quad" and "double" indicate the employed internal precision.

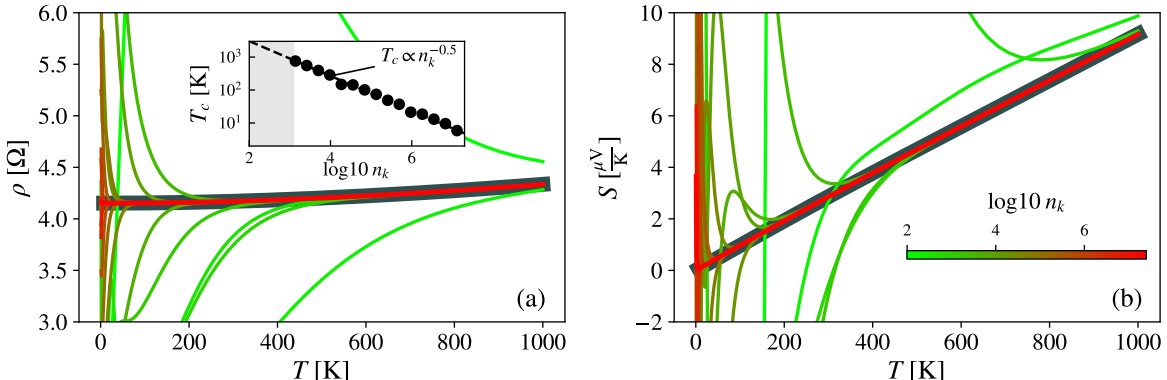

Figure 7: (a) Resistivity and (b) Seebeck coefficient for the two-dimensional electronic structure of Eq. (80) with $\Gamma = 10^{-4}$eV, system charge $N = 1.2$ and various numbers of **k**-points $n_k$. The convergence temperature $T_c$ (inset in (a)) scales as $T_c \propto n_k^{-0.5} = n_{k_x}^{-1}$ (fitted, dashed line). A dense momentum grid is required to reach convergence (fat gray line) for reasonably low temperatures. There, we observe an almost temperature independent resistivity and a linear-in-$T$ behavior of the Seebeck coefficient. The identical behavior is observed in the three dimensional equivalent where instead the convergence temperature scales as $T_c \propto n_k^{-0.33} = n_{k_x}^{-1}$ (not shown).

## 5 Test cases

In this section we are going to apply the previously described transport methodology. Our aim is to highlight some of the features implemented in LINRETRACE on systems that cover a wide range of phenomena, without, however, exhausting the code's full functionality. We start off with simple models in Sec. 5.1, Sec. 5.2 and Sec. 5.3 and transition to realistic crystal structures in Sec. 5.4 and Sec. 5.5. Input and configuration files of all examples can be found

in the github.com/linretrace/linretrace_examples repository.[22]

## 5.1    One-band metal in two dimensions

We consider the simple electronic structure

$$\epsilon^0_{\mathbf{k}n} = -2t \sum_{\alpha=x,y} \cos(k_\alpha) \tag{80}$$

with hopping amplitude $t = 0.25$eV, lattice spacing $a_x = a_y = 1$Å and optical elements determined with the Peierls approximation. In order to introduce particle-hole asymmetry (to generate a finite thermopower) we set the system's charge to $N = 1.2$ and determine the chemical potential with the digamma occupation Eq. (70). Figure 7 shows the conductivity and Seebeck coefficient for a temperature-independent scattering rate $\Gamma = 10^{-4}$eV, no quasi-particle renormalization $Z = 1$, and various momentum grids. Once $\mathbf{k}$-convergence is reached, the conductivity is essentially temperature independent. Then, as expected from the Mott formula [79], a linear-in-$T$ behaviour appears in the Seebeck coefficient. As the system is above half-filling, the carriers are of hole-type, and the Seebeck coefficient is positive. Unsurprisingly, a dense momentum grid is required to reach convergence at low temperatures. For the largest momentum grid employed here ($n_k = n_{k_x} \times n_{k_y} = 5000 \times 5000$) the results are converged down to approximately $T_c = 6$K. Please note that due to the chemical potential search, discrepancies for coarse $\mathbf{k}$-meshes are the results of a mixture of chemical potential and kernel sampling errors. In all, even if the $\mathbf{k}$-convergence in LinReTraCe is more stable than in Boltzmann codes, a thorough check of the Brillouin-zone discretization is *mandatory*—at least when the response is metallic.

## 5.2    Two-band insulator in three dimensions

Next, we consider the two-band electronic structure used in Ref. [2]

$$\epsilon^0_{\mathbf{k}n} = - \sum_{\alpha=x,y,z} 2t_n \cos(k_\alpha) + (-1)^n (6t_n + \Delta_0/2) \tag{81}$$

with valence band hopping amplitude $t_1 = 0.25$eV and lattice spacing $a_x = a_y = a_z = 1$Å. Here, we use the Peierls approach in the band-basis, see Eq. (55), to compute matrix elements, limiting the response to intra-band transitions. For the temperature scan we use a conduction band hopping $t_2 = -0.30$eV and a band gap of $\Delta_0 = 0.1$eV while for the chemical potential scan we use $t_2 = -0.25$eV and $\Delta_0 = 1$eV. On the calculated temperature range, full momentum-grid convergence is achieved for $60 \times 60 \times 60$ $\mathbf{k}$-points. Contrary to the metallic system, the transport coefficients of this insulator *must* be evaluated with quadruple precision for $\Gamma > 0$ in order to avoid numeric instabilities in the polygamma functions at low temperatures. Instead, evaluating the Boltzmann kernels, a much denser momentum grid, that scales similarly to the metallic system of Sec. 5.1 is required for convergence (not shown). Therefore, the more proper treatment of scattering amplitudes in our formalism actually facilitates the numerical evaluation with respect to semi-classical approaches.

---

[22]For storage reasons, the hosted HDF5 input files contain data on a coarser momentum mesh than that used for the results shown here. For the model systems, however, the user can directly reproduce our results by generating their own HDF5 input files using the number of $\mathbf{k}$-points specified in the text. The same is possible for the material test cases, but requires performing ones own DFT calculations.

### 5.2.1 Temperature scan

The introduced band asymmetry leads to an imbalance of electronic carriers as the chemical potential lies above $\mu = 0$, see bottom panel of Fig. 8a. The thermal activation across the charge gap is reflected in all considered physical observables where one naturally finds $\rho, R_H \propto e^{-\frac{\Delta_0}{2k_B T}}$ and $S, \nu \propto \frac{1}{T}$ at high enough temperatures. At low enough temperatures, however, a saturation regime is observed in $\rho$ and $R_H$, stemming from the Kubo transport kernels, see Ref. [2] for details. Further, entropy transport is thermodynamically consistent, as $S \propto T$ and $\nu \propto T$ towards absolute zero. Taking lifetime broadening into account also for the occupation, the chemical potential $\mu_\psi(T)$ is necessarily (since $|t_2| > |t_1|$) forced towards the conduction band from which a second activated regime is generated below 100K, see Fig. 8a (top panel). This *intrinsic* chemical potential behavior can also be achieved, in principle, through a guidance of $\mu_{\mathrm{FD}}(T)$ via *explicit* impurity states, see Fig. 8b. Here, sharper transitions between regimes and even changes in the dominant type of carriers can be generated. Please note the different plotting scales and the sign changes of the Seebeck and Hall coefficient.

### 5.2.2 Chemical potential scan

The chemical potential scans in Fig. 9 illustrate the in-gap behavior of the (intra-band) conductivity and the (intra-band) Seebeck coefficient. For small scattering rates and elevated temperatures (Fig. 9a), we recover the usual 'S'-shaped curve for the Seebeck coefficient: $S$ crosses zero at $\mu = 0$, where the system is particle-hole symmetric. Slightly above (below) a minimum (maximum) develops, corresponding to the dominant type of carriers ($\mu > 0$: electrons; $\mu < 0$: holes). In the 'Boltzmann' regime we find perfect agreement with the Goldsmid rule $2eS_{\max}T = \Delta_0$ [80], relating the maximal Seebeck coefficient $S_{\max}$ to the system's gap. At lower temperatures, however, deviations from this behavior can be observed as a plateau around $\mu = 0$ develops that expands as we lower the temperature further. This effect stems from the Kubo kernels Eqs. (23-28) and signals the transition from the activated to a saturation regime. For a detailed discussion, see Ref. [2]. Larger scattering rates (Fig. 9b) already lead to a deviation from the 'Boltzmann' behavior at the highest temperatures, where we additionally find a strong suppression towards the band edges at $\mu = \pm 0.5$eV.

### 5.3 Honeycomb lattice (tight-binding calculation)

To showcase the post-processing capabilities of LinReTraCe we revisit the honeycomb structure of Sec. 4.1: We define the lattice vectors of length 1Å via

$$a_1 = \begin{pmatrix} 1 \\ 0 \end{pmatrix}; \; a_2 = \frac{1}{2}\begin{pmatrix} 1 \\ \sqrt{3} \end{pmatrix} \tag{82}$$

resulting in the reciprocal lattice vectors

$$b_1 = \frac{2\pi}{\sqrt{3}}\begin{pmatrix} \sqrt{3} \\ -1 \end{pmatrix}; \; b_2 = \frac{4\pi}{\sqrt{3}}\begin{pmatrix} 0 \\ 1 \end{pmatrix}, \tag{83}$$

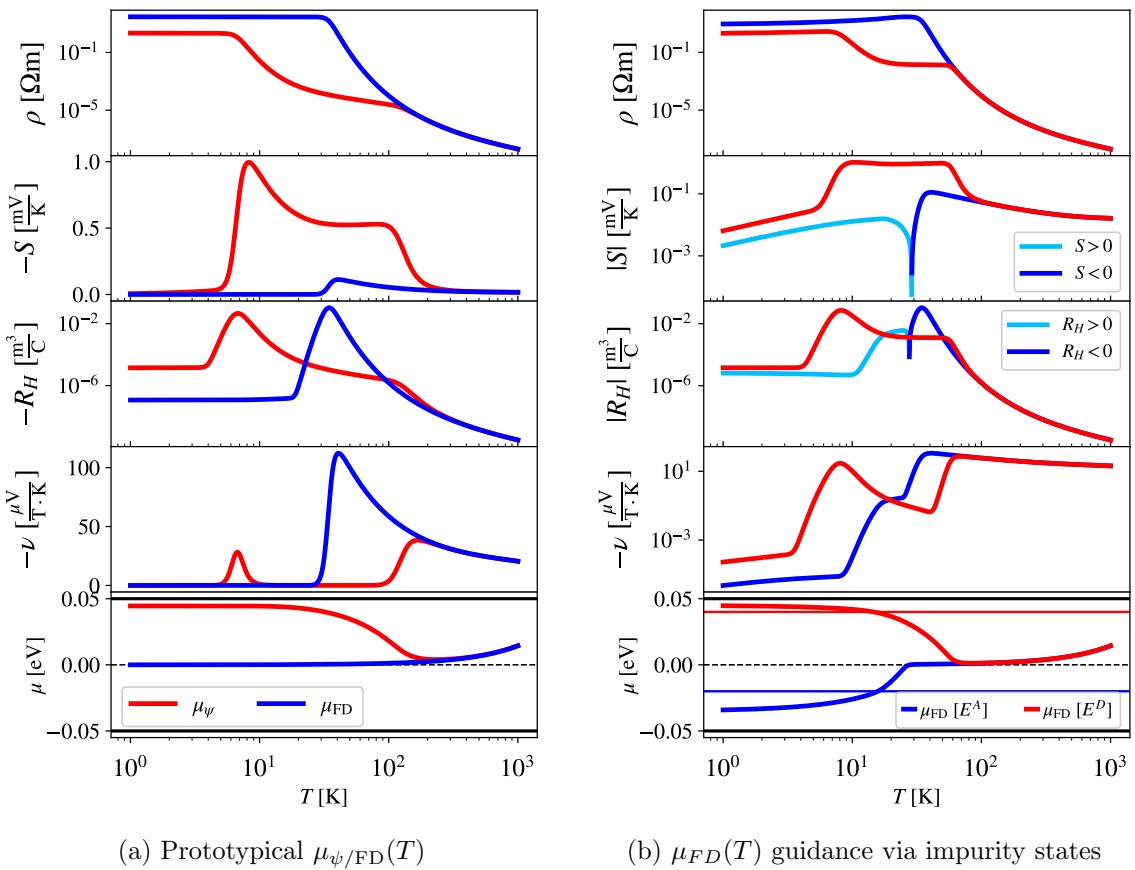

(a) Prototypical $\mu_{\psi/\mathrm{FD}}(T)$     (b) $\mu_{FD}(T)$ guidance via impurity states

Figure 8: Prototypical temperature dependencies in a semiconductor of the resistivity $\rho$ (top), the coefficients of Seebeck $S$, Hall $R_H$, and Nernst $\nu$ (second to fourth panel), as well as the chemical potential (bottom; the band gap is delimited by thick black lines): (a) Effects stemming purely from the occupation distribution (Fermi-Dirac "FD" or digamma "$\psi$"). (b) Effects achievable by guiding the chemical potential with impurity states. Red: donor level $E^D = 0.04$eV, $\rho_D = 10^{-8}\frac{1}{\text{unit cell}}$; Blue: acceptor level $E^A = -0.02$eV, $\rho_A = 10^{-14}\frac{1}{\text{unit cell}}$. In all cases $\Delta_0 = 100$meV, $\Gamma = 10^{-4}$eV; $Z = 1$.

fulfilling $a_i \cdot b_j = 2\pi\delta_{i,j}$. The special points K and K' (see Fig. 11) are then located at

$$K = \frac{2}{3}b_1 + \frac{1}{3}b_2 \tag{84}$$

$$K' = \frac{1}{3}b_1 + \frac{2}{3}b_2 \tag{85}$$

hence momentum grids that are divisible by three in each direction are required to include them. We choose $n_{k_x} \times n_{k_y} = 300 \times 300$ with nearest neighbor hopping $t_{AB} = t_{BA} = 1$eV. The corresponding tight binding file[23] reads:

---

[23]This honeycomb structure, among other structures, including kagome, 3D bcc, 3D fcc, etc., is available in the code repository's `templates` subfolder.

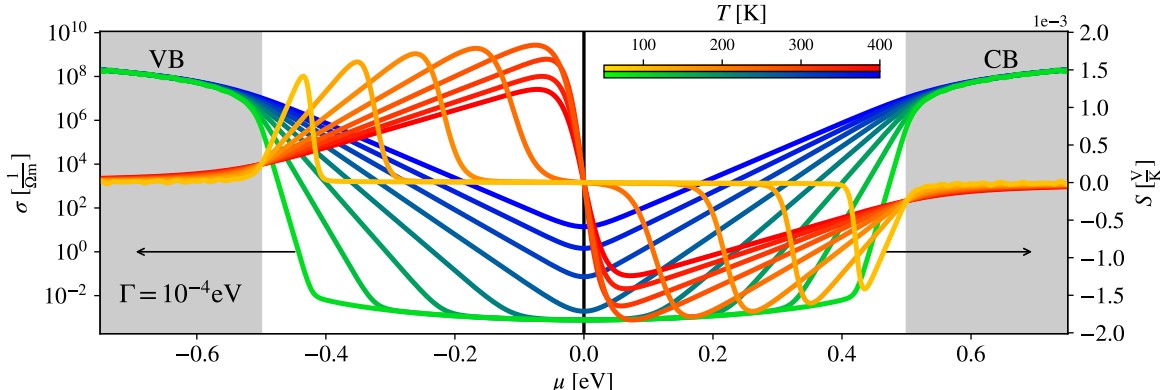

(a) $\sigma(\mu)$ and $S(\mu)$ for various temperatures with $\Gamma = 10^{-4}$eV.

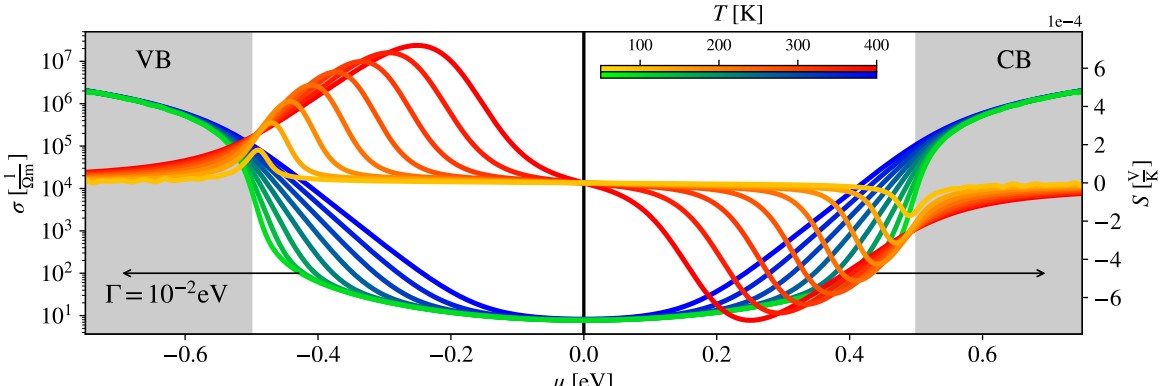

(b) $\sigma(\mu)$ and $S(\mu)$ for various temperatures with $\Gamma = 10^{-2}$eV.

Figure 9: Chemical potential scans for the symmetric insulator $t_1 = -t_2 = 0.25$ eV for (a) small $\Gamma = 10^{-4}$eV and (b) large $\Gamma = 10^{-2}$eV scattering rates. Blue-to-green shades represent conductivities; red-to-orange shades Seebeck coefficients. In the Boltzmann temperature regime ($\Gamma = 10^{-4}$eV; $T > 250$K) we find perfect agreement with the Goldsmid rule $2eS_{\max}T = \Delta_0$. Lower temperatures and increased scattering leads to a departure of the Boltzmann regime and the transport responses qualitatively change shape. The white (grey shaded) background indicates the gap (valence/conduction band region). The gap is $\Delta_0 = 1$eV and we used $Z = 1$.

```
begin hopping
   0   0   0      1 2 1.0
   0   0   0      2 1 1.0
  +1   0   0      2 1 1.0
   0  +1   0      2 1 1.0
  -1   0   0      1 2 1.0
   0  -1   0      1 2 1.0
end hopping

begin atoms
  1 0.25 0.25 0
  1 0.75 0.75 0
end atoms

begin real_lattice
  1    0               0
  0.5 0.8660254038 0
  0    0               1
end real_lattice
```

The resulting band-structure and density of states are illustrated in Fig. 10. The particle-hole symmetry around $\varepsilon = 0$ results in symmetric transport properties $\sigma_{\alpha\beta}(+\mu) = \sigma_{\alpha\beta}(-\mu)$. Here, we consider $\mu = 0.8\text{eV}$, as indicated by the horizontal dashed line. Fig. 11 (left) shows the momentum-resolved optical elements $M^{\alpha\beta} \propto v_\alpha v_\beta$ in and around the Brillouin zone. Combined with the kernel function $\mathcal{K}_{11}$ (middle), we can capture how each point in the Brillouin zone contributes to the conductivity $\sigma_{\alpha\beta} = \sum_{\mathbf{k}\in\text{BZ}} \sigma_{\alpha\beta}(\mathbf{k})$ (right panel). Via an interplay of symmetry, the sign, and the values of the optical elements we find the expected result: $\sigma_{xx} = \sigma_{yy}$ and $\sigma_{xy} \equiv 0$. This type of analysis can be easily extended to other transport coefficients and models and can provide valuable microscopic insight.

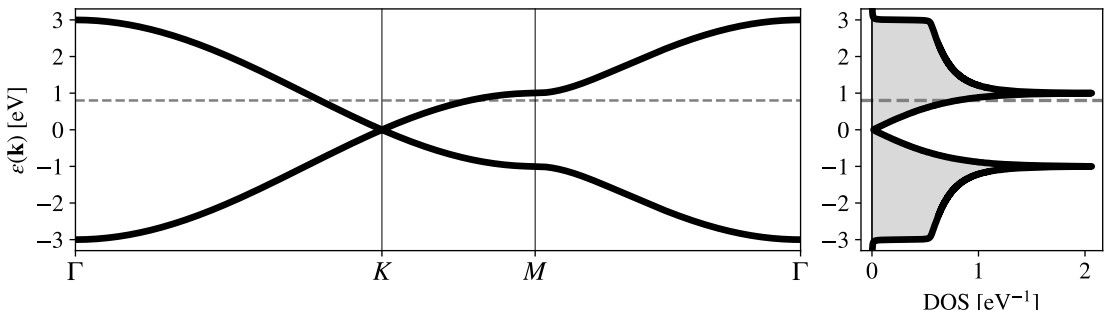

Figure 10: Band-structure of the honeycomb lattice (left) and the density of states (right). The analysis of Fig. 11 is done for $\mu = 0.8$eV (dashed line).

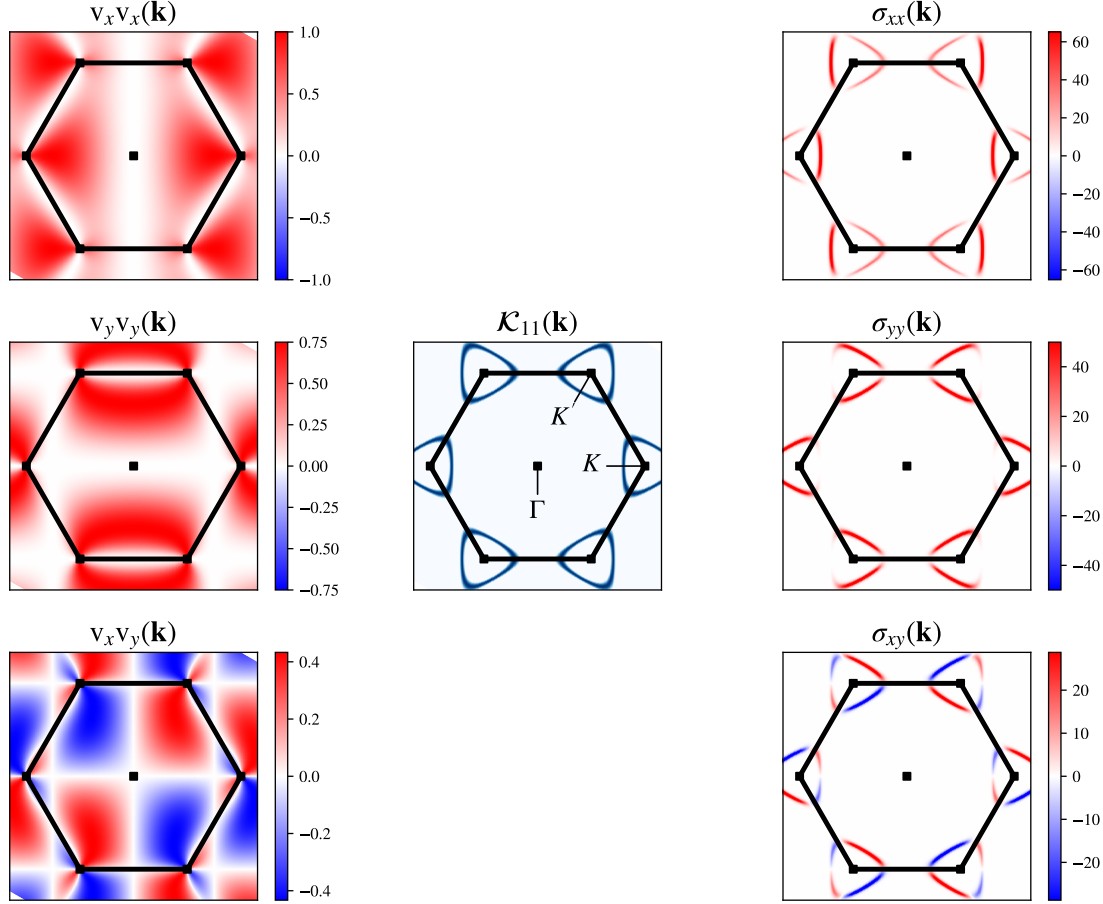

Figure 11: Momentum analysis of the honeycomb lattice. Left: Optical elements in units of $[\text{eV}^2\text{Å}^2]$ for the $xx$ (top), $yy$ (middle) and $xy$ (bottom) polarizations. Middle: Kernel function $\mathcal{K}_{11}$, reflecting the Fermi surface at $\mu = 0.8$eV. Right: Momentum-resolved conductivity $\sigma(\mathbf{k})$ in units of $[\frac{1}{\Omega}]$ which is the left and middle column multiplied with each other. $\sum_{\mathbf{k}} \sigma(\mathbf{k})$ results in $\sigma_{xx} = \sigma_{yy}$ and $\sigma_{xy} \equiv 0$.

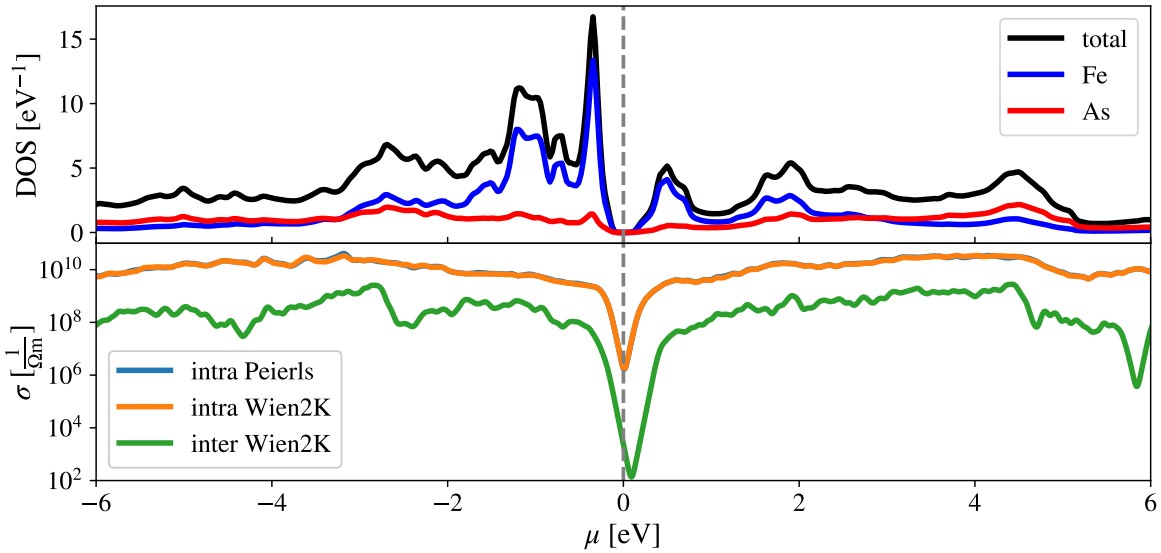

Figure 12: Hybridization-gap semiconductor FeAs$_2$. Result of a `WIEN2K` calculation using the PBE exchange-correlation potential and $20 \times 20 \times 40$ **k**-points. The DOS around the gap ($\Delta = 0.275$eV) is dominated by Fe-character. (top) total and partial density of states (DOS), (bottom) intra- and inter-band conductivities in the $x$-direction and as a function of the chemical potential $\mu$. The Peierls approximation to the optical elements leads to intra-band results (blue) virtually indistinguishable to using the full dipole matrix elements (yellow). For the given temperature ($T = 300$K) and scattering rate ($\Gamma = 10^{-5}$eV) the intra-band contributions largely dominate over inter-band effects (green).

## 5.4    FeAs$_2$ (WIEN2k calculation)

As a realistic crystal structure we first consider FeAs$_2$, a hybridization-gap semiconductor. Band-theory yields a gap of around $\Delta_0 = 0.275$eV when using the PBE exchange-correlation potential [4,81]. The top panel of Fig. 12 shows the ensuing density of states (DOS) for a wide energy range, with the gap centered at $\mu = 0$. The states near the gap edges are dominantly of iron character. The bottom panel shows the resulting conductivities $\sigma$ calculated for $T = 300$K, using a scattering rate $\Gamma_0 = 10^{-5}$eV. Comparing the intra-band conductivity using the `WIEN2K` dipole matrix elements (yellow) with that employing group velocities constructed from the `BOLTZTRAP2` band interpolation (blue) illustrates the accuracy of the Peierls approximation for simple unit cells.[24] With the full dipole matrix element, we have access also to inter-band contributions to the conductivity (green). As expected, inter-band contributions only play a subsidiary role at these elevated temperatures: They are two orders of magnitude smaller than the intra-band contributions.

Translating the $\mu$-axis into carrier concentrations $n = (N(\mu) - N)/V$ results in the be-

---

[24]We cross-checked our result with `BoltzTraP`: Differences are small. We note that `BoltzTraP2` calculates transport distribution functions that are based on an artificially broadened density of states, leading to a numerical smearing of the gap edges. At its core, `LINRETRACE` is designed to stay numerically exact and avoids unphysical broadening of any transport quantity. If desired, however, the user can apply a Gaussian broadening of a chemical potential scan in the post-processing. For explicit comparisons, the `BoltzTraP2` conductivities $\sigma/\tau_0 \; [\frac{1}{\Omega \text{ms}}]$ have to be multiplied with $\tau_0 = \frac{\hbar}{2\Gamma_0}$, $\hbar = 6.58211956 \cdot 10^{-16}$eVs.

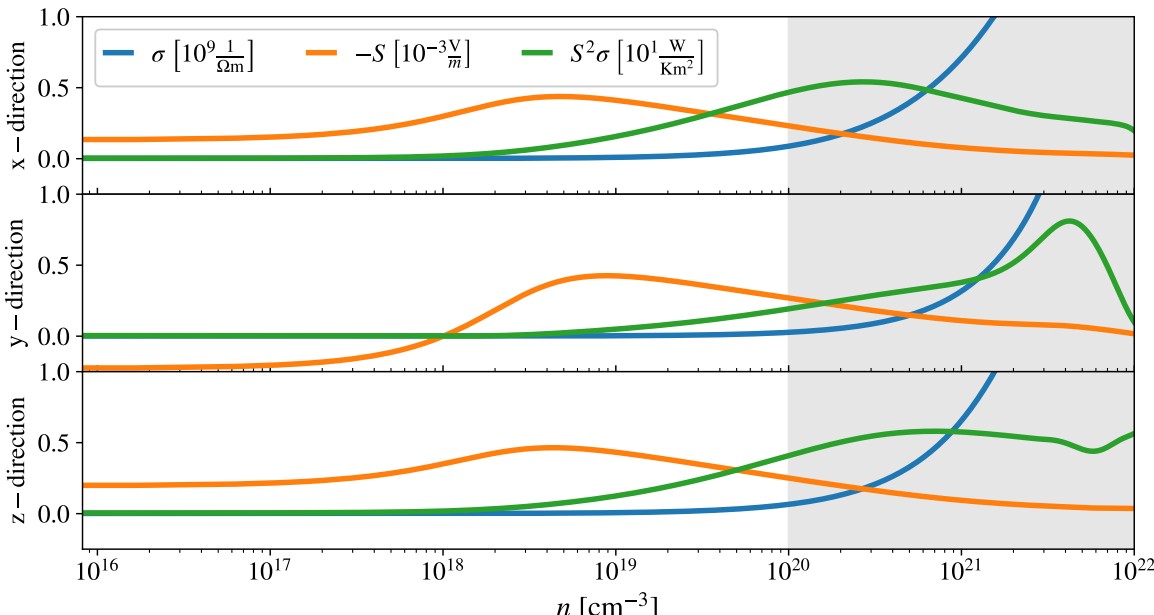

Figure 13: FeAs$_2$: WIEN2K intra-band conductivity ($\sigma$), Seebeck coefficient ($S$) and power factor ($S^2\sigma$) for the same temperature ($T = 300$K) and scattering rate ($\Gamma = 10^{-5}$eV) as in Fig. 12. Here, we show the dependency on the electron doping $n = \frac{N(\mu) - N_{\text{neutral}}}{V}$, instead of the chemical potential $\mu$. The three Cartesian directions ($x$, $y$, $z$) refer to the orthorhombic unit-cell with volume $V = a_x \times a_y \times a_z = 5.30\text{Å} \times 5.98\text{Å} \times 2.88\text{Å}$. The gray, shaded area indicates doping levels where the chemical potential has moved inside the conduction band.

havior shown in Fig. 13. For the given temperature, the shaded gray area marks the region, where the chemical potential has moved inside the conduction band. Upon entering said region, the conductivity increases rapidly, while the Seebeck coefficient has its peak amplitude for $\mu$ inside the gap, cf. Figure 9. Indeed, the behavior of $\sigma$ and $S$ is typically [82], but not always [83], antagonistic. The response is polarization dependent, with the powerfactor $S^2\sigma$ being most sensitive on the crystal orientation.[25,26]

Finally, we add an in-gap impurity state and showcase the ensuing temperature behavior of transport observables in comparison to experiment [62, 89] in Fig. 14. Due to the additional electrons provided by the donor level, located in the vicinity of the conduction band, the system experiences a transition from activated behavior with the intrinsic gap $\Delta_0$ at high temperatures to a second regime controlled by a reduced gap $\Delta_1$, determined by the position of the impurity, cf. Fig. 4 and Ref. [2]. At still lower temperatures, and akin to Sec. 5.2, the prototypical saturation regimes set in: The resistivity and the Hall coefficient saturate, while the Seebeck and Nernst coefficients tend towards zero in a linear fashion. Due to the

---

[25]Please note that these result use the full dipole matrix element, for which WIEN2K has a maximal number of computable **k**-points. Results for the metallic regime could therefore not be checked for momentum-convergence. This limitation does not apply for Peierls velocities.

[26]LINRETRACE only computes thermoelectric properties from pure electron diffusion. Phonon-drag enhancements—relevant, e.g., for the related narrow-gap semiconductors FeSb$_2$ [84–86] and CrSb$_2$ [87,88]—are not included.

three distinct crystal directions in FeAs$_2$, the Hall and Nernst coefficients behave differently depending on the direction of the applied magnetic field.

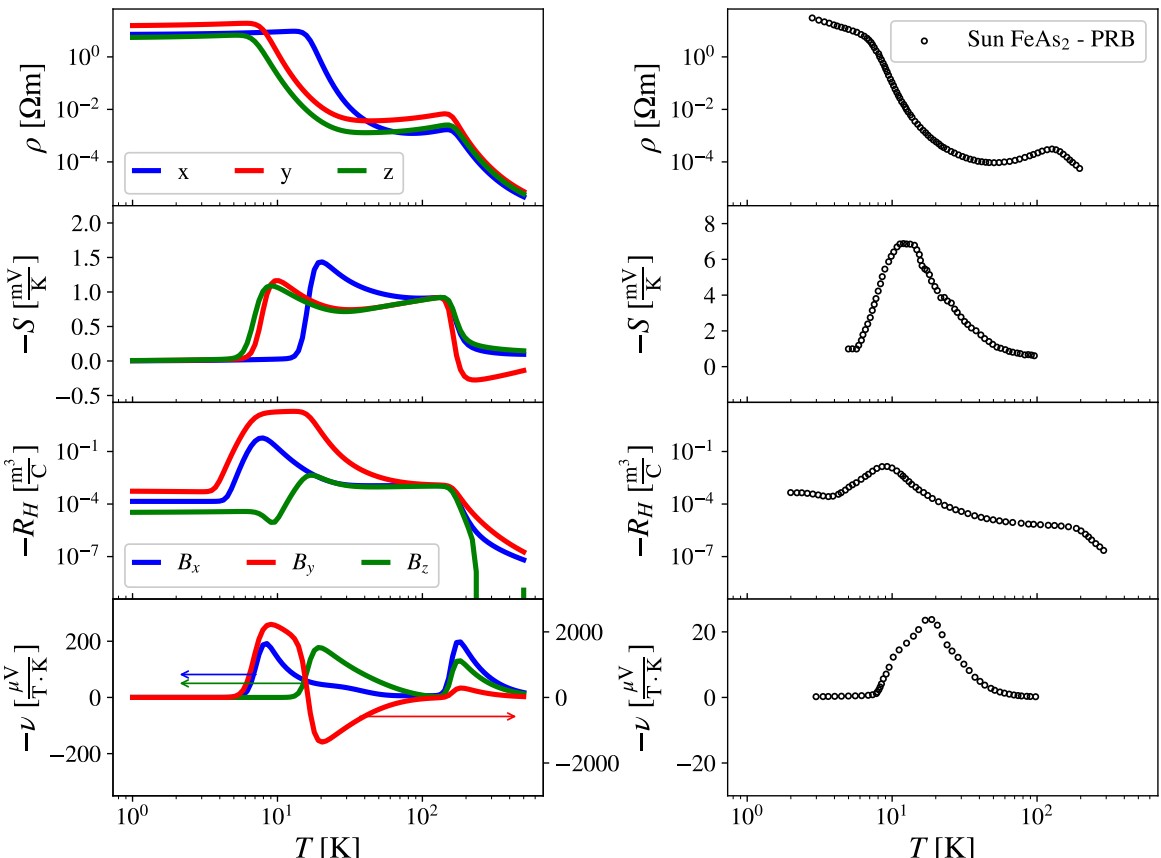

Figure 14: FeAs$_2$: Intra-band resistivity $\rho$, Seebeck coefficient $S$, Hall coefficient $R_H$ and Nernst coefficient $\nu$ (left) compared to experimental data (right) [62]. We show results for all three Cartesian directions. The chemical potential was determined by the Fermi Dirac function and altered by a donor level (at $E_D = 0.015$eV below the conduction band with $\rho = 2 \cdot 10^{-6} \frac{1}{\text{unit cell}}$), chosen so that the resistivity approximates the experimental behavior. The scattering rate includes only polynomial terms: $\Gamma(T) = 7 \cdot 10^{-5} + 5 \cdot 10^{-9} T^2$ in eV. Note that we underestimate $S$ and overestimate $\nu$. All shown data use the intra-band Peierls approximation.

## 5.5   Tl-doped PbTe (WIEN2k calculation)

As a final test material we consider Tl-doped PbTe, a prime example for the enhancement of thermoelectric transport by resonance states [90]. We model the doping by explicitly constructing a $4 \times 4 \times 4$ supercell of PbTe and substituting a single Pb atom with a Tl one. Internal positions of the resulting Tl$_{0.004}$Pb$_{0.996}$Te are then fully relaxed before extracting the LinReTraCe input. In order to gain access to both the Seebeck and the Nernst coefficient, we employ the BoltzTraP2 interpolation scheme to obtain Peierls velocities. As the prepared super cell does not match the experimental stoichiometry, we are mainly aiming for qualitative

aspects.

Through the Tl-(hole)doping an increase of the density of states in the vicinity of the valence-band edge appears, see Figure 15 and the emerging resonance pins the chemical potential. Consequently, the Tl-doping cause a semiconductor-to-metal transition. Supple-

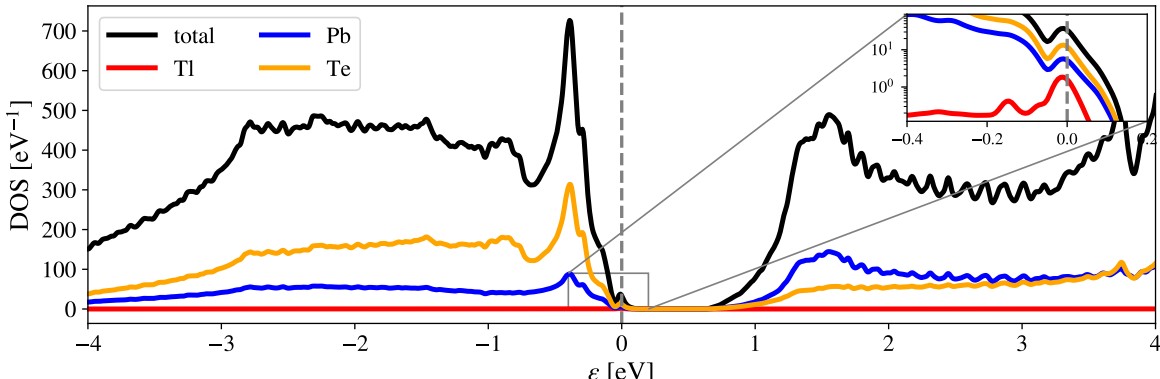

Figure 15: Tl-doped PbTe: Density of states using the LDA exchange-correlation potential and $17 \times 17 \times 17$ **k**-points. Introducing a single Tl$\Longleftrightarrow$Pb substitutional impurity results in a distorted electronic structure and additional states appear near the valence-band edge, pinning the chemical potential. The zoomed inset illustrates the non-trivial nature of the resonance to which all three elements contribute in some part.

menting the electronic structure with a phenomenological, band-independent $\Gamma(T) = 20$meV $+6 \cdot 10^{-5}$meV/K$^2 T^2$ results in the resistivity, Seebeck, and Nernst coefficient shown in Fig. 16. Comparing to experimental results [90], plotted on the same scale, we find good agreement for the resistivity at elevated temperature. However, we do not capture the resonance's signature below 200K. This finding suggests that the resistivity in that temperature range is not controlled by band-structure effects. Instead, the scattering rate might be markedly different for the resonance than for other states and display a more complex temperature dependence. We hence expect that—with scattering rates obtained from more sophisticated (beyond band-theory) electronic structure methodologies—LinReTraCe could capture the resistivity also below 200K. Alternatively, a reverse engineering approach could be employed within LinReTraCe to *extract* a phenomenological but band-dependent scattering rate that reproduces the experimental resistivity.

The Seebeck and Nernst coefficient, instead, are well reproduced within our setting, without any additional input, suggesting a simple electronic picture of thermoelectric transport to hold. Indeed, in the case of metals, the Seebeck coefficient is—to a first approximation—insensitive to the scattering rate. The congruence to experiment for $S$ then supports the above claim that the temperature profile of the resistivity below 200K is controlled by an intricate scattering rate.

The metallic nature of the transport poses the previously discussed challenges for the Brillouin-zone discretization. While the resistivity and the Seebeck coefficient are, for all practical purposes, convergent with the largest **k**-mesh used, the Nernst coefficient is notably more sensitive: The shown data provides a good approximation for the Nernst coefficient above 200K, while below the result is clearly not yet converged. Indeed, for general reasons,

the Nernst coefficient must vanish for $T \to 0$ [2]. We stress that these limitations regarding the number of usable **k**-points are on the side of the electronic structure methodology, while LinReTraCe could handle larger meshs at acceptable costs.

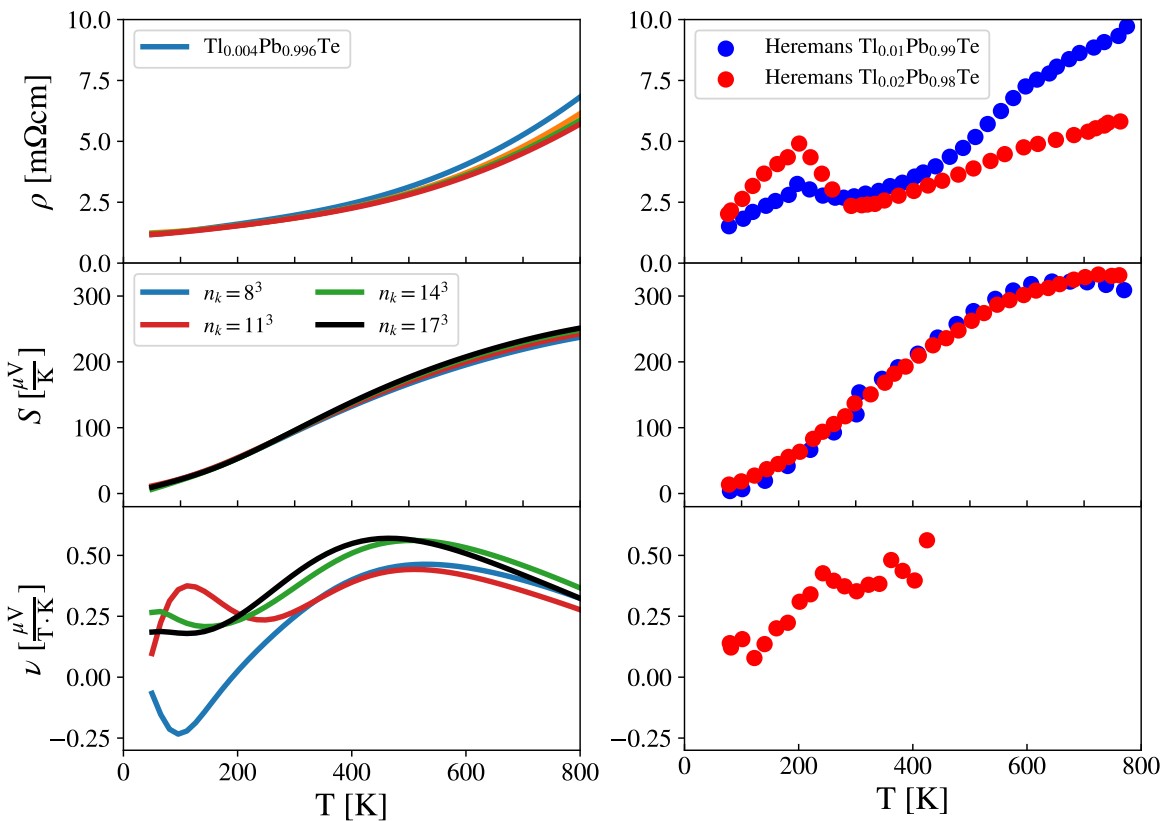

Figure 16: Tl-doped PbTe: Resistivity (top) and coefficients of Seebeck (middle) and Nernst (bottom) within LinReTraCe (left) compared to experiment [90] (right). Even though the resonance feature of the resistivity below 200K cannot be reproduced with our phenomenological scattering rate, the Seebeck and Nernst coefficients are matched nearly one-to-one. Indeed, while the resistivity is directly controlled by our ansatz, $\Gamma(T) = 20\text{meV} + 6 \cdot 10^{-5}\text{meV/K}^2 T^2$, for the scattering, the Seebeck coefficient in metals is (to first approximation) independent of $\Gamma$. There are opposing conventions for the sign of the Nernst coefficient [61]. Since our Hall signal (not shown) has the same sign as the experiment [90], but not the Nernst coefficient, we multiply the experimental $\nu$ with $(-1)$. Please also note that $\nu \to 0$ for $T \to 0$ is a must. Indeed, due to the limited number of **k**-points, the Nernst simulation is not fully converged below $T = 200$K.

# 6 Conclusion

We presented the algorithm, implementation, and usage of LinReTraCe, a package for the computation of transport properties of solids. The code's strengths are (1) based on Kubo

linear-response theory it captures effects of finite lifetimes far beyond semi-classical theories, making a qualitative difference in semi-conductors; (2) using a semi-analytical evaluation of transport kernels, LinReTraCe is nonetheless as fast as Boltzmann approaches in the relaxation-time approximation; (3) LinReTraCe is agnostic to the origin of the input data, allowing to simulate tight-binding models as well as materials. Interfaces to WIEN2k, VASP, and WANNIER90 are included in the release, while a template allows connecting LinReTraCe to any other electronic structure code. The numerical efficiency makes LinReTraCe also attractive for high-throughput material surveys of, e.g., thermoelectrics.

## Acknowledgements

The authors acknowledge discussions with A. Georges, J. Mitscherling, and J. Mravlje. Calculations were performed on the Vienna Scientific Cluster (VSC).

**Author contributions**   M.P. and E.M. developed the methodology and the program package. M.P. and J.M.T. created the manuscript. J.M.T. designed and supervised the research.

**Funding information**   LinReTraCe was funded by the Austrian Science Fund (FWF) through project P 30213.

# A   Cheat sheet

Code repository:
https://github.com/linretrace/linretrace.
User guide:
https://github.com/linretrace/linretrace/raw/release/documentation/userguide.pdf.
Create an energy file from `WIEN2k`, `VASP`, `WANNIER90`, a tight binding file or generic data:

```
$ ldft <wien2k folder> --optic      # wien2k with full dipole elements
$ ldft <vasp folder> --interp       # vasp with intra-band interpolation
$ lwann <wannier90 folder>          # wannier90
$ ltb <tb file> <nkx> <nky> <nkz> <charge>   # tight-binding input
$ linterface                        # your customized interface
```

The full capability of the Python interfaces can be shown via:

```
$ ldft/lwann/ltb --help
```

The created energy files can be inspected with `lprint`:

```
$ lprint    <lrtc input> info
$ lprint -p <lrtc input> dos
$ lprint -p <lrtc input> path
```

*Optional:* Prepare a scattering file by copying `lscat_temp` into your working directory and adjust it to your needs.
Next, prepare a basic config file with

```
$ lconfig
```

Advanced option can be added with the help of the full configuration specifications found in `documentation/configspec`.
Then, perform LinReTraCe calculation via:

```
$ mpirun -np <cores> bin/linretrace config.lrtc
```

When finished, the datasets contained in the output can be inspected via:

```
$ lprint <lrtc output> list        # physical datasets
$ lprint <lrtc output> olist       # Onsager coefficients
```

Printing or plotting (-p option) the data:

```
$ lprint -p latest mu --gap        # chemical potential + gap
$ lprint -p latest c-intra uxx dxx # conductivity
$ lprint -p latest r-intra xx      # resistivity
$ lprint -p latest p-intra xy      # Peltier
$ lprint -p latest s-intra xx yy zz # Seebeck
$ lprint -p latest rh-intra xyz    # Hall
$ lprint -p latest n-intra xyz     # Nernst
$ lprint -p latest L12B-intra uxyz # L12B Onsager
```

# B  Advanced `LinReTraCe` configuration

Section 3.5 provides a minimal configuration for temperature and chemical potential runs. Here, we detail more advanced capabilities of LINRETRACE. A complete documentation of the config specifications can be found in `documentation/configspec`.

## B.1  General settings

In the `[General]` section of the config file

```
[General]
Bandgap           = <value>
ElectronOccupation = <value>
FermiOccupation    = T/F         # default = T
FullOutput         = <string>
# <string> = full -> momentum resolved, band resolved
# <string> = ksum -> momentum summed,   band resolved
# <string> = bsum -> momentum resolved, band summed
```

one can manipulate the band gap, change the total occupation, and activate the output to be fully or partially resolved into momentum and band contributions. The band gap can only be changed if a gap has been detected in the pre-processing. Hence a properly set occupation in these setups is necessary. The change in the band gap is achieved via an upwards shift of all the conduction bands, while the valence bands remain fixed. For spin-polarized systems, two values need to be provided, one for each spin.

The change in total occupation via the config file triggers a re-search of the 'DFT' chemical potential (having a direct impact on chemical-potential run, that are defined with respect to the updated $\mu_{\mathrm{DFT}}$).

The used occupation function can be switched between the digamma function $f(\varepsilon) = \frac{1}{\pi} - \Im\psi(\frac{1}{2} + \frac{\beta}{2\pi}(\Gamma + ia))$ and the Fermi function $f(\varepsilon) = [1 + e^{\beta a}]^{-1}$ via the `FermiOccupation` key word. Please note that this affects the chemical potential as well as all quantities depending on it, including the carrier concentration, total energy, etc.

The `FullOutput` option can be configured to either output the fully resolved (`full`), or partially resolved (momentum sum: `ksum`, band sum: `bsum`) Onsager coefficients. The tree structure of the full dependence output is (in contrast to the standard output) separated for each step, otherwise only the dataset shapes and their identifier differ

```
/step/000001              Group
/step/000001/L11          Group
/step/000001/L11/intra    Group
/step/000001/L11/intra/full  Dataset #    FullOutput = full
/step/000001/L11/intra/ksum  Dataset # or FullOutput = ksum
/step/000001/L11/intra/bsum  Dataset # or FullOutput = bsum
...
```

where the datasets without magnetic field have the array shape `[nkp,spins,bands,3,3]` and the datasets with magnetic field are of shape `[nkp,spins,bands,3,3,3]`. Partially momentum (band) summed datasets have `nkp = 1` (`bands = 1`). The steps are always ordered the

same way as the summed quantities, that is from lowest to highest temperature or from lowest to highest chemical potential, independent on how the program itself loops over them. These datasets can only be accessed via external HDF5 libraries, an exemplary load in via `h5py` looks like

```
import h5py
with h5py.File('lrtc-output.hdf5','r') as h5:
    L11full  = h5['step/000001/L11/intra/full'][()]
    L11Bfull = h5['step/000001/L11B/intra/full'][()]
print(L11full.shape)  # kpoints, spins, bands, 3, 3
print(L11Bfull.shape) # kpoints, spins, bands, 3, 3, 3
```

## B.2   Temperature mode settings

The temperature sub configuration includes the following options:

```
[TempMode]
ConstantMu       = <value>
OldMuHdf5        = <LRTCoutput file>
OldMuText        = <lprint mu output>
NImp             = <value>
Doping           = <value>
NominalDoping    = T/F                 # default = F
```

`ConstantMu` sets the chemical potential to a fixed absolute value (in eV) with respect to $\mu_{\mathrm{DFT}}$ for the full temperature run. `OldMuHdf5` uses the chemical potential values of an old output file. `OldMuText` can be used to enter manual values via a text file whose format has to be consistent with the output of `lprint <LRTCoutput file> mu` (column 1 temperature, column 2: chemical potential). Naturally these three option cannot be used in combination.

Impurity levels and/or homogeneous doping, as illustrated in Sec. 4.2.1 and Sec. 4.2.2, respectively, can be activated via `Doping` and `NImp`. The value in `Doping` is in units of $[\mathrm{cm}^{-3}]$ which can be changed to nominal doping [1/unit cell] by setting `NominalDoping = T`. This behavior then also applies to the introduced $n$ impurity states, activated via `NImp = n`, for which further configuration is required:

```
[[Impurities]]
[[[1]]]
# one of
Absolute   = <values>
Valence    = <values>
Conduction = <values>
Percentage = <values>

Bandtype   = <string>
# <string> = box, triangle, halfcircle, sine, sine2, sine3, sine4
Bandwidth  = <value>
[[[2]]]
...
```

Separate groups `[[[1]]]` ... have to be introduced to describe each of the $n$ impurities. The main descriptor is *one* of the four keywords `Absolute`, `Valence`, `Conduction` or `Percentage`, as they describe how the provided energy is to be interpreted. The provided string then contains 4 values, that describe the type of impurity (donor: $+1$/acceptor $-1$), its density, energy, and degeneracy. The energy is translated into an internal position, depending on the keyword: `Absolute` uses the energy is as, `Valence` interprets the energy as positive offset with respect to the top of the valence band, `Conduction` as a negative offset with respect to the bottom of the conduction band. `Percentage` puts the impurity state at the fractional position inside the gap ($0 < E_{\text{Percentage}} < 1$). Naturally the latter three options can only be used in gapped systems. Energies refer to bare values, prior to potential renormalizations with $Z$ and $\Re\Sigma$. Example inputs for impurity states look as follows

```
# Identifier = type density energy degeneracy
Absolute   = -1   1e15    8.35   2    # acceptor at E = 8.35 eV
Conduction = +1   1e18    0.02   1    # donor 20meV below CB
Valence    = -1   1e19    0.01   1    # acceptor 10meV above VB
Percentage = +1   1e15    0.5    2    # donor at center of gap
```

These states can be transformed into bands by providing the `Bandtype` identifier (cf. Section 4.2.1) and an associated `Bandwidth`. The introduced bandwidth is centered around the provided energy and has the form of the provided `Bandtype` string. All these impurity states only affect the chemical potential; they do not contribute to the transport.

## B.3  Scattering files

Runs with a provided scattering file are configured with

```
[TempMode] # or [MuMode]
[[Scattering]]
ScatteringFile    = <scattering file>
ScatteringOffset  = <value>
```

as the scattering file already contains the defining calculation axis. Here we allow for an additional offset value that is added on top of the provided scattering dependencies, i.e. $\Gamma(k, n, \mu|T) + \Gamma_{\text{offset}}$, e.g., to mimic additional impurity scattering. This subsection can be used in the temperature, as well as the chemical potential sub configuration. However we cross-check the internally set calculation mode of the scattering file with the calculation mode of the config file.

# C   Dimensional analysis

## C.1  Internal units

`LinReTraCe` works internally with energies in [eV] and distances (momenta) in [Å] ([Å$^{-1}$]). The Onsager coefficients in Eqs. (3-4) are represented as transport kernels $\mathcal{K}$ and velocities as matrix elements $\mathcal{M}$. To transform the optical elements into our internal units, $\hbar-$factors

need to be moved to the pre-factor:

$$\mathcal{L}_{ab}^{\alpha\beta} = \frac{\pi e^{(4-a-b)}}{\hbar V} \sum_{\substack{n,m \\ \mathbf{k},\sigma}} \mathcal{K}_{ab}(\mathbf{k}, n, m) \overline{M}^{\alpha\beta}(\mathbf{k}, n, m) \tag{86}$$

$$\mathcal{L}_{ab}^{B,\alpha\beta\gamma} = \frac{4\pi^2 e^{(5-a-b)}}{3\hbar^2 V} \sum_{\substack{n,m \\ \mathbf{k},\sigma}} \mathcal{K}_{ab}^B(\mathbf{k}, n, m) \overline{M}^{B,\alpha\beta\gamma}(\mathbf{k}, n, m) \tag{87}$$

With this transformation the units of the 'new' optical elements are

$$\overline{M}^{\alpha\beta} = \hbar^2 M^{\alpha\beta} = \left[ \mathrm{eV}^2 \mathring{\mathrm{A}}^2 \right] \tag{88}$$

$$\overline{M}^{B,\alpha\beta\gamma} = \hbar^3 M^{B,\alpha\beta\gamma} = \left[ \mathrm{eV}^3 \mathring{\mathrm{A}}^4 \right] . \tag{89}$$

The kernels with $a, b \in \{1, 2\}$ naturally are

$$\mathcal{K}_{ab} = \left[ \mathrm{eV}^{(a+b-4)} \right] \tag{90}$$

$$\mathcal{K}_{ab}^B = \left[ \mathrm{eV}^{(a+b-5)} \right] \tag{91}$$

if one uses scattering rates $\Gamma$ and energies $a$ in [eV] as well as inverse temperatures $\beta = \frac{1}{k_B T}$ in [eV$^{-1}$]. Assuming a three-dimensional unit cell of volume $V \left[ \mathring{\mathrm{A}}^3 \right]$, we combine the previous units to obtain the Onsager coefficients. Additional scaling related post-factors are required to generate SI units:

$$\mathcal{L}_{11} = \underbrace{\frac{\pi e^2 [\mathrm{C}^2]}{\hbar [\mathrm{Js}] \, V [\mathring{\mathrm{A}}^3]}}_{pre} \mathcal{K}_{11} [\mathrm{eV}^{-2}] \, \overline{M}^{\alpha\beta} [\mathrm{eV}^2 \mathring{\mathrm{A}}^2] \underbrace{10^{10} \left[ \frac{\mathring{\mathrm{A}}}{\mathrm{m}} \right]}_{post} \tag{92}$$

$$\rightarrow \left[ \frac{\mathrm{C}^2}{\mathrm{Js} \mathring{\mathrm{A}}^3} \mathrm{eV}^{-2} \mathrm{eV}^2 \mathring{\mathrm{A}}^2 \frac{\mathring{\mathrm{A}}}{\mathrm{m}} \right] = \left[ \frac{\mathrm{A}}{\mathrm{Vm}} \right]$$

$$\mathcal{L}_{12} = \underbrace{\frac{\pi e [\mathrm{C}]}{\hbar [\mathrm{Js}] \, V [\mathring{\mathrm{A}}^3]}}_{pre} \mathcal{K}_{12} [\mathrm{eV}^{-1}] \, \overline{M}^{\alpha\beta} [\mathrm{eV}^2 \mathring{\mathrm{A}}^2] \underbrace{10^{10} \left[ \frac{\mathring{\mathrm{A}}}{\mathrm{m}} \right] e \left[ \frac{\mathrm{J}}{\mathrm{eV}} \right]}_{post} \tag{93}$$

$$\rightarrow \left[ \frac{\mathrm{C}}{\mathrm{Js} \mathring{\mathrm{A}}^3} \mathrm{eV}^{-1} \mathrm{eV}^2 \mathring{\mathrm{A}}^2 \frac{\mathring{\mathrm{A}}}{\mathrm{m}} \frac{\mathrm{J}}{\mathrm{eV}} \right] = \left[ \frac{\mathrm{A}}{\mathrm{m}} \right]$$

$$\mathcal{L}_{22} = \underbrace{\frac{\pi}{\hbar [\mathrm{Js}] \, V [\mathring{\mathrm{A}}^3]}}_{pre} \mathcal{K}_{12} [\mathrm{eV}^0] \, \overline{M}^{\alpha\beta} [\mathrm{eV}^2 \mathring{\mathrm{A}}^2] \underbrace{10^{10} \left[ \frac{\mathring{\mathrm{A}}}{\mathrm{m}} \right] e^2 \left[ \frac{\mathrm{J}^2}{\mathrm{eV}^2} \right]}_{post} \tag{94}$$

$$\rightarrow \left[ \frac{1}{\mathrm{Js} \mathring{\mathrm{A}}^3} \mathrm{eV}^2 \mathring{\mathrm{A}}^2 \frac{\mathring{\mathrm{A}}}{\mathrm{m}} \frac{\mathrm{J}^2}{\mathrm{eV}^2} \right] = \left[ \frac{\mathrm{VA}}{\mathrm{m}} \right]$$

$$\mathcal{L}^B_{11} = \underbrace{\frac{4\pi^2 e[\mathrm{C}]^3}{3\hbar^2 [\mathrm{J^2 s^2}]\, V[\mathrm{\mathring{A}}^3]}}_{pre} \mathcal{K}^B_{11}[\mathrm{eV}^{-3}]\, \overline{M}^{B,\alpha\beta\gamma}[\mathrm{eV^3 \mathring{A}}^4]\, \underbrace{10^{-10}\left[\frac{\mathrm{m}}{\mathrm{\mathring{A}}}\right]}_{post} \tag{95}$$

$$\rightarrow \left[\frac{\mathrm{C}^3}{\mathrm{J^2 s^2 \mathring{A}}^3}\mathrm{eV}^{-3}\mathrm{eV^3 \mathring{A}}^4\frac{\mathrm{m}}{\mathrm{\mathring{A}}}\right] = \left[\frac{\mathrm{Am}}{\mathrm{V^2 s}}\right]$$

$$\mathcal{L}^B_{12} = \underbrace{\frac{4\pi^2 e[\mathrm{C}]^2}{3\hbar^2 [\mathrm{J^2 s^2}]\, V[\mathrm{\mathring{A}}^3]}}_{pre} \mathcal{K}^B_{12}[\mathrm{eV}^{-2}]\, \overline{M}^{B,\alpha\beta\gamma}[\mathrm{eV^3 \mathring{A}}^4]\, \underbrace{10^{-10}\left[\frac{\mathrm{m}}{\mathrm{\mathring{A}}}\right] e\left[\frac{\mathrm{J}}{\mathrm{eV}}\right]}_{post} \tag{96}$$

$$\rightarrow \left[\frac{\mathrm{C}^2}{\mathrm{J^2 s^2 \mathring{A}}^3}\mathrm{eV}^{-2}\mathrm{eV^3 \mathring{A}}^4\frac{\mathrm{m}}{\mathrm{\mathring{A}}}\frac{\mathrm{J}}{\mathrm{eV}}\right] = \left[\frac{\mathrm{Am}}{\mathrm{Vs}}\right]$$

$$\mathcal{L}^B_{22} = \underbrace{\frac{4\pi^2 e[\mathrm{C}]}{3\hbar^2 [\mathrm{J^2 s^2}]\, V[\mathrm{\mathring{A}}^3]}}_{pre} \mathcal{K}^B_{22}[\mathrm{eV}^{-1}]\, \overline{M}^{B,\alpha\beta\gamma}[\mathrm{eV^3 \mathring{A}}^4]\, \underbrace{10^{-10}\left[\frac{\mathrm{m}}{\mathrm{\mathring{A}}}\right] e^2\left[\frac{\mathrm{J}^2}{\mathrm{eV}^2}\right]}_{post} \tag{97}$$

$$\rightarrow \left[\frac{\mathrm{C}}{\mathrm{J^2 s^2 \mathring{A}}^3}\mathrm{eV}^{-1}\mathrm{eV^3 \mathring{A}}^4\frac{\mathrm{m}}{\mathrm{\mathring{A}}}\frac{\mathrm{J}^2}{\mathrm{eV}^2}\right] = \left[\frac{\mathrm{Am}}{\mathrm{s}}\right]$$

The magnetic and non-magnetic Onsager coefficients are hence connected unit-wise via the magnetic field $B[\mathrm{T}]$

$$\mathcal{L}_{ab} = \mathcal{L}^B_{ab}\left[\frac{\mathrm{Vs}}{\mathrm{m}^2}\right]. \tag{98}$$

Naturally, the current densities in Eqs. (1-2) result in

$$j^\alpha_e = \mathcal{L}^{\alpha\beta}_{11} E^\beta - \frac{1}{T}\mathcal{L}^{\alpha\beta}_{12}\partial_\beta T = \left[\frac{\mathrm{A}}{\mathrm{m}^2}\right] \tag{99}$$

$$j^\alpha_q = -\mathcal{L}^{\alpha\beta}_{21} E^\beta - \frac{1}{T}\mathcal{L}^{\alpha\beta}_{22}\partial_\beta T = \left[\frac{\mathrm{VA}}{\mathrm{m}^2}\right] = \left[\frac{\mathrm{W}}{\mathrm{m}^2}\right]. \tag{100}$$

Combined to the physical observables listed in Eqs. (11-20) we obtain

$$\sigma_{\alpha\beta} = \left[\frac{A}{Vm}\right] = \left[\frac{1}{\Omega m}\right] \tag{101}$$

$$\rho_{\alpha\beta} = \left[\frac{Vm}{A}\right] = [\Omega m] \tag{102}$$

$$S_{\alpha\beta} = \left[\frac{V}{K}\right] \tag{103}$$

$$\kappa_{\alpha\beta} = \left[\frac{VA}{Km}\right] = \left[\frac{W}{Km}\right] \tag{104}$$

$$\sigma^B_{\alpha\beta\gamma} = \left[\frac{Am}{V^2}\right] \tag{105}$$

$$R_{H,\alpha\beta\gamma} = \left[\frac{m^3}{As}\right] = \left[\frac{m^3}{C}\right] \tag{106}$$

$$\nu_{\alpha\beta\gamma} = \left[\frac{m^2}{sK}\right] = \left[\frac{V}{TK}\right] \tag{107}$$

$$\mu_{H,\alpha\beta\gamma} = \left[\frac{m^2}{Vs}\right] = \left[\frac{1}{T}\right] \tag{108}$$

$$\mu_{T,\alpha\beta\gamma} = \left[\frac{m^2}{Vs}\right] = \left[\frac{1}{T}\right] \tag{109}$$

## C.2   External units

WIEN2k works internally in energy units of [Ry] and length units of [Bohr]:

$$V_{\mathrm{LRTC}}\left[\mathring{A}^3\right] = V_{\mathrm{WIEN2k}}\left[\mathrm{Bohr}^3\right] \times 0.529177^3 \left[\frac{\mathring{A}^3}{\mathrm{Bohr}^3}\right] \tag{110}$$

$$\varepsilon_{\mathrm{LRTC}}\left[\mathrm{eV}\right] = \varepsilon_{\mathrm{WIEN2k}}\left[\mathrm{Ry}\right] \times 13.605662285 \left[\frac{\mathrm{eV}}{\mathrm{Ry}}\right] \tag{111}$$

The optical elements are essentially dipole moments $\langle n, \mathbf{k}|\partial_i|n'\mathbf{k}\rangle$, whose units are $\left[\mathrm{Bohr}^{-1}\right]$. First, the derivatives are transformed into velocities ($\partial_i \to p = \hbar\partial_i \to v = \frac{p}{m_e}$), before being transformed according to Eq. (88).

$$\left[\mathrm{Bohr}^{-1}\right] \times \frac{1}{0.529177\left[\frac{\mathring{A}}{\mathrm{Bohr}}\right]} \times 10^{10}\left[\frac{\mathring{A}}{m}\right] \to \left[m^{-1}\right] \tag{112}$$

$$\left[m^{-1}\right] \times \frac{\hbar\left[\mathrm{Js}\right]}{m_e\left[\mathrm{kg}\right]} \to \left[\frac{m}{s}\right] \tag{113}$$

$$\left[\frac{m}{s}\right] \times \frac{\hbar\left[\mathrm{Js}\right]}{e\left[\frac{J}{\mathrm{eV}}\right]} \times 10^{10}\left[\frac{\mathring{A}}{m}\right] \to \left[\mathrm{eV}\mathring{A}\right] \tag{114}$$

Since WIEN2k outputs symmetrized squared dipole moments the final transformation looks as

follows

$$
\overline{M}_{\text{LRTC}} \left[\text{eV}^2 \text{Å}^2\right] = M_{\text{WIEN2k}} \left[\text{Bohr}^{-2}\right] \times \underbrace{\left( \frac{\hbar^2 [\text{J}^2\text{s}^2] \, 10^{20} \left[\frac{\text{Å}^2}{\text{m}^2}\right]}{0.529177 \left[\frac{\text{Å}}{\text{Bohr}}\right] \, e \left[\frac{\text{J}}{\text{eV}}\right] \, m_e[\text{kg}]} \right)^2}_{\approx 207.35}
\tag{115}
$$

**VASP** already works in units of [eV] and [Å], so no transformations are necessary.
**BoltzTraP2** works in energy units of [Ha] and length units of [Bohr]. The transformed energies and derivatives follow accordingly

$$
\varepsilon_{\text{LRTC}} \left[\text{eV}\right] = \varepsilon_{\text{BTP2}} \left[\text{Ha}\right] \times 27.21132457 \left[\frac{\text{eV}}{\text{Ha}}\right]
\tag{116}
$$

$$
\partial_{\mathbf{k}} \varepsilon_{\text{LRTC}} \left[\text{eVÅ}\right] = \partial_{\mathbf{k}} \varepsilon_{\text{BTP2}} \left[\text{Ha Bohr}\right] \times 27.21132457 \left[\frac{\text{eV}}{\text{Ha}}\right] \times 0.529177 \left[\frac{\text{Å}}{\text{Bohr}}\right]
\tag{117}
$$

$$
\partial_{\mathbf{k}}^2 \varepsilon_{\text{LRTC}} \left[\text{eVÅ}^2\right] = \partial_{\mathbf{k}}^2 \varepsilon_{\text{BTP2}} \left[\text{Ha Bohr}^2\right] \times 27.21132457 \left[\frac{\text{eV}}{\text{Ha}}\right] \times 0.529177^2 \left[\frac{\text{Å}^2}{\text{Bohr}^2}\right]
\tag{118}
$$

With the latter two, the optical elements in the Peierls approximation follow

$$
\overline{M}_{\text{LRTC}}^{\alpha\beta} \left[\text{eV}^2 \text{Å}^2\right] = \partial_{\mathbf{k}_\alpha} \varepsilon \left[\text{eVÅ}\right] \partial_{\mathbf{k}_\beta} \varepsilon \left[\text{eVÅ}\right]
\tag{119}
$$

$$
\overline{M}_{\text{LRTC}}^{B,\alpha\beta\gamma} \left[\text{eV}^3 \text{Å}^4\right] = \varepsilon_{\gamma ij} \partial_{\mathbf{k}_\alpha} \varepsilon \left[\text{eVÅ}\right] \partial_{\mathbf{k}_\beta} \partial_{\mathbf{k}_i} \varepsilon \left[\text{eVÅ}^2\right] \partial_{\mathbf{k}_j} \varepsilon \left[\text{eVÅ}\right] .
\tag{120}
$$

## C.3    Lower dimensionality

The units derived above rely on a three-dimensional unit cell. If one performs calculations for lower dimensional models or non-periodic structures the volume definition necessarily changes.

### C.3.1    two dimensions

In pure 2D (models) and quasi 2D systems (non-periodic crystal structures, e.g., thin films) no properly defined lattice constant exists in the non-periodic direction. The 'volume' thus changes from $\left[\text{Å}^3\right]$ to $\left[\text{Å}^2\right]$. In these cases the Onsager coefficients have to be *manually* adapted *a posteriori* by multiplying with the employed lattice constant (in [m]) in the non-periodic direction. Subsequently the units of the Onsager coefficients change, resulting in changes to the conductivity (resistivity), thermal conductivity, and the Hall coefficient. As the other physical observables consist of ratios of equal number of coefficients, their values and units are unaffected.

$$\sigma^{2D} = \left[\frac{1}{\Omega}\right] \tag{121}$$

$$\rho^{2D} = [\Omega] \tag{122}$$

$$\kappa^{2D} = \left[\frac{W}{K}\right] \tag{123}$$

$$R_H^{2D} = \left[\frac{m^2}{C}\right] \tag{124}$$

As the two-dimensional resistivity (sheet resistance) could be misinterpreted as a resistance one tends instead to use the equivalent notation $\rho^{2D} = [\Omega/\square]$ in this context.

### C.3.2    one dimension

In the same way, one-dimensional systems have to be adapted. Contrary to the two dimensional case, however, no magnetic quantities can be defined as there do not exist three distinct directions required for the Hall and Nernst coefficient.

$$\sigma^{1D} \propto \left[\frac{m}{\Omega}\right] \tag{125}$$

$$\rho^{1D} \propto \left[\frac{\Omega}{m}\right] \tag{126}$$

$$\kappa^{1D} \propto \left[\frac{Wm}{K}\right] \tag{127}$$

## D    Interband - intraband limit

When evaluating the inter-band kernels Eqs. (29-31) one has to do thorough checks of the involved energies $a_{1/2}$ and scattering rates $\Gamma_{1/2}$. If these parameters are too close to each other in the complex plane, the numerical evaluation may become unstable. Then, the proper analytic intra-band limit has to be taken instead. This commonly occurs in calculations with band-crossings when band- and momentum-independent scattering rates are employed.

In this Section we illustrate that the inter-band derived formulas are indeed consistent and taking the degenerate limit result in the intra-band expressions. First, we construct the vectorial difference of the poles in the upper half of the complex plane stemming from the spectral functions of Eq. 9:

$$\bar{z} = (a_2 + i\Gamma_2) - (a_1 + i\Gamma_1) \equiv Re^{i\Phi} \tag{128}$$

We can now express the '2' variables via the '1' variables

$$a_2 = a_1 + R\cos(\Phi) \tag{129}$$

$$\Gamma_2 = \Gamma_1 + R\sin(\Phi) \tag{130}$$

$$\psi_1(z_2) = \psi_1\left(\frac{1}{2} + \frac{\beta}{2\pi}\left(\Gamma_1 + ia_1\right) + \frac{\beta}{2\pi}R\sin(\Phi) + i\frac{\beta}{2\pi}R\cos(\Phi)\right). \tag{131}$$

and expand Eq. (131) around $R = 0$:

$$
\psi_1 \left( \frac{1}{2} + \frac{\beta}{2\pi} \left( \Gamma_2 + i a_2 \right) \right) =
$$
$$
\psi_1 \left( \frac{1}{2} + \frac{\beta}{2\pi} \left( \Gamma_1 + i a_1 \right) \right) + \psi_2 \left( \frac{1}{2} + \frac{\beta}{2\pi} \left( \Gamma_1 + i a_1 \right) \right) \frac{\beta R}{2\pi} \left( \sin(\Phi) + i \cos(\Phi) \right) + \mathcal{O}(R^2)
$$

(132)

Setting $Z_1 = Z_2 = Z$, $a_1 = a$, $\Gamma_1 = \Gamma$, and inserting the above equations in the $\mathcal{K}_{11}$ inter-band expression results in

$$
\mathcal{K}_{11}(\mathbf{k}, n, m) = \frac{Z^2 \beta}{2\pi^3 R^2 \left[ R^2 + 4\Gamma^2 + 4\Gamma R \sin(\Phi) \right]}
$$
$$
\times \Bigg[ \Re \Big\{ \left[ R^2 + 2\Gamma R \sin(\Phi) - 2i\Gamma R \cos(\Phi) \right] \left( \Gamma + R \sin(\Phi) \right) \psi_1(z) \Big\}
$$
$$
+ \Re \Big\{ \left[ R^2 (\cos^2(\Phi) - \sin^2(\Phi)) - 2\Gamma R \sin(\Phi) + 2i\Gamma R \cos(\Phi) + 2i R^2 \sin(\Phi) \cos(\Phi) \right] \Gamma
$$
$$
\times \left[ \psi_1(z) + \psi_2(z) \frac{\beta R}{2\pi} \left( \sin(\Phi) + i \cos(\Phi) \right) + \mathcal{O}(R^2) \right] \Big\} \Bigg].
$$

(133)

To lowest order, the pre-factor term scales with $\mathcal{O}(R^{-2})$, so in order to recover all non-vanishing terms we have to check the square bracket for terms up to $\mathcal{O}(R^2)$:

$$
\mathcal{K}_{11}(\mathbf{k}, n, m) = \left[ \frac{Z^2 \beta}{8\pi^3 R^2 \Gamma^2} + \mathcal{O}(R^{-1}) \right]
$$
$$
\times \Bigg[ R \times \underbrace{\Big\{ 2\Gamma^2 \sin(\Phi) \Re \psi_1(z) + 2\Gamma^2 \cos(\Phi) \Im \psi_1(z) - 2\Gamma^2 \sin(\Phi) \Re \psi_1(z) - 2\Gamma^2 \cos(\Phi) \Im \psi_1(z) \Big\}}_{\equiv 0}
$$
$$
+ R^2 \times \Big\{ \Gamma \Re \psi_1(z) + 2\Gamma \sin^2(\Phi) \Re \psi_1(z) + 2\Gamma \cos(\Phi) \sin(\Phi) \Im \psi_1(z)
$$
$$
+ (\cos^2(\Phi) - \sin^2(\Phi)) \Gamma \Re \psi_1(z) - 2 \sin(\Phi) \cos(\Phi) \Gamma \Im \psi_1(z)
$$
$$
+ -2\Gamma^2 \sin^2(\Phi) \frac{\beta}{2\pi} \Re \psi_2(z) + 2\Gamma^2 \sin(\Phi) \cos(\Phi) \frac{\beta}{2\pi} \Im \psi_2(z)
$$
$$
- 2\Gamma^2 \cos(\Phi) \sin(\Phi) \frac{\beta}{2\pi} \Im \psi_2(z) - 2\Gamma^2 \cos^2(\Phi) \frac{\beta}{2\pi} \Re \psi_2(z) \Big\} + \mathcal{O}(R^3) \Bigg]
$$

(134)

As expected, the terms scaling with $\mathcal{O}(R)$ cancel exactly while the $\mathcal{O}(R^2)$ terms neatly recover

the intra-band expression

$$\mathcal{K}_{11}(\mathbf{k}, n, n) = \lim_{R \to 0^+} \mathcal{K}_{11}(\mathbf{k}, n, m) = \frac{Z^2 \beta}{8\pi^3 \Gamma^2}$$

$$\times \left[ \Re\psi_1(z) \underbrace{\left[\Gamma + 2\Gamma \sin^2(\Phi) + \Gamma(\cos^2(\Phi) - \sin^2(\Phi))\right]}_{\equiv 2\Gamma} + \Im\psi_1(z) \underbrace{\left[2\Gamma \cos(\Phi)\sin(\Phi) - 2\Gamma \sin(\Phi)\cos(\Phi)\right]}_{\equiv 0} \right.$$

$$\left. + \Re\psi_2(z) \underbrace{\left[-2\Gamma^2 \sin^2(\Phi)\frac{\beta}{2\pi} - 2\Gamma^2 \cos^2(\Phi)\frac{\beta}{2\pi}\right]}_{\equiv -\frac{\Gamma^2 \beta}{\pi}} + \Im\psi_2(z) \underbrace{\left[2\Gamma^2 \sin(\Phi)\cos(\Phi)\frac{\beta}{2\pi} - 2\Gamma^2 \cos(\Phi)\sin(\Phi)\frac{\beta}{2\pi}\right]}_{\equiv 0} \right].$$

$$(135)$$

$$\mathcal{K}_{11}(\mathbf{k}, n, n) = \frac{Z^2 \beta}{4\pi^3 \Gamma} \left[ \Re\psi_1(z) - \frac{\beta\Gamma}{2\pi} \Re\psi_2(z) \right] \qquad (136)$$

The other two kernels can be derived in a similar fashion:

$$\mathcal{K}_{12}(\mathbf{k}, n, m) = \left[ \frac{Z^2 \beta}{8\pi^3 R^2 \Gamma^2} + \mathcal{O}(R^{-1}) \right]$$

$$\times \left[ \Re\left\{ (a - i\Gamma) \left[R^2 + 2\Gamma R \sin(\Phi) - 2i\Gamma R \cos(\Phi)\right] (\Gamma + R\sin(\Phi))\psi_1(z) \right\} \right.$$

$$+ \Re\left\{ (a + R\cos(\Phi) - i\Gamma - iR\sin(\Phi)) \right.$$

$$\times \left[ R^2(\cos^2(\Phi) - \sin^2(\Phi)) - 2\Gamma R\sin(\Phi) + 2i\Gamma R\cos(\Phi) + 2iR^2 \sin(\Phi)\cos(\Phi)\right] \Gamma$$

$$\left. \left. \times \left[ \psi_1(z) + \psi_2(z)\frac{\beta R}{2\pi}(\sin(\Phi) + i\cos(\Phi)) + \mathcal{O}(R^2)\right] \right\} \right]$$

$$(137)$$

$$\mathcal{K}_{22}(\mathbf{k}, n, m) = \left[ \frac{Z^2 \beta}{8\pi^3 R^2 \Gamma^2} + \mathcal{O}(R^{-1}) \right]$$

$$\times \left[ \Re\left\{ (a - i\Gamma)^2 \left[R^2 + 2\Gamma R \sin(\Phi) - 2i\Gamma R \cos(\Phi)\right] (\Gamma + R\sin(\Phi))\psi_1(z) \right\} \right.$$

$$+ \Re\left\{ (a + R\cos(\Phi) - i\Gamma - iR\sin(\Phi))^2 \right.$$

$$\times \left[ R^2(\cos^2(\Phi) - \sin^2(\Phi)) - 2\Gamma R\sin(\Phi) + 2i\Gamma R\cos(\Phi) + 2iR^2 \sin(\Phi)\cos(\Phi)\right] \Gamma$$

$$\left. \left. \times \left[ \psi_1(z) + \psi_2(z)\frac{\beta R}{2\pi}(\sin(\Phi) + i\cos(\Phi)) + \mathcal{O}(R^2)\right] \right\} \right]$$

$$(138)$$

To avoid unnecessarily lengthy expressions from here on, we restrict ourselves to $\Phi = 0$:

$$\mathcal{K}_{12}(\mathbf{k}, n, m) = \left[ \frac{Z^2\beta}{8\pi^3 R^2 \Gamma^2} + \mathcal{O}(R^{-1}) \right]$$
$$\times \left[ \Re\left\{ (a - i\Gamma) \left[R^2 - 2i\Gamma R\right] \Gamma \psi_1(z) \right\} \right.$$
$$\left. + \Re\left\{ (a + R - i\Gamma) \left[R^2 + 2i\Gamma R\right] \Gamma \left[ \psi_1(z) + \psi_2(z) \frac{i\beta R}{2\pi} + \mathcal{O}(R^2) \right] \right\} \right] \tag{139}$$

$$\mathcal{K}_{22}(\mathbf{k}, n, m) = \left[ \frac{Z^2\beta}{8\pi^3 R^2 \Gamma^2} + \mathcal{O}(R^{-1}) \right]$$
$$\times \left[ \Re\left\{ (a - i\Gamma)^2 \left[R^2 - 2i\Gamma R\right] \Gamma \psi_1(z) \right\} \right.$$
$$\left. + \Re\left\{ (a + R - i\Gamma)^2 \left[R^2 + 2i\Gamma R\right] \Gamma \left[ \psi_1(z) + \psi_2(z) \frac{i\beta R}{2\pi} + \mathcal{O}(R^2) \right] \right\} \right] \tag{140}$$

Expanding all terms, we again see that those scaling with $R$ cancel exactly,

$$\mathcal{K}_{12}(\mathbf{k}, n, m) = \left[ \frac{Z^2\beta}{8\pi^3 R^2 \Gamma^2} + \mathcal{O}(R^{-1}) \right]$$
$$\times \left[ R^2 \times \left\{ a\Gamma\Re\psi_1(z) + \Gamma^2\Im\psi_1(z) + a\Gamma\Re\psi_1(z) - 2a\Gamma^2\frac{\beta}{2\pi}\Re\psi_2(z) - 2\Gamma^2\Im\psi_1(z) \right. \right.$$
$$\left. \left. + \Gamma^2\Im\psi_1(z) - 2\Gamma^3\frac{\beta}{2\pi}\Im\psi_2(z) \right\} + \mathcal{O}(R^3) \right] \tag{141}$$

$$\mathcal{K}_{22}(\mathbf{k}, n, m) = \left[ \frac{Z^2\beta}{8\pi^3 R^2 \Gamma^2} + \mathcal{O}(R^{-1}) \right]$$
$$\times \left[ R^2 \times \left\{ (a^2 - \Gamma^2)\Gamma\Re\psi_1(z) + 2a\Gamma^2\Im\psi_1(z) + (a^2 - \Gamma^2)\Gamma\Re\psi_1(z) + 2a\Gamma^2\Im\psi_1(z) \right. \right.$$
$$\left. \left. - (a^2 - \Gamma^2)2\Gamma^2\frac{\beta}{2\pi}\Re\psi_2(z) - 4a\Gamma^3\frac{\beta}{2\pi}\Im\psi_2(z) - 4a\Gamma^2\Im\psi_1(z) + 4\Gamma^3\Re\psi_1(z) \right\} + \mathcal{O}(R^3) \right], \tag{142}$$

and taking the $R \to 0$ limit recovers the intra-band expressions

$$\mathcal{K}_{12}(\mathbf{k}, n, n) = \frac{Z^2\beta}{4\pi^3\Gamma} \left[ a\Re\psi_1(z) - a\frac{\beta\Gamma}{2\pi}\Re\psi_2(z) - \frac{\beta\Gamma^2}{2\pi}\Im\psi_2(z) \right] \tag{143}$$

$$\mathcal{K}_{22}(\mathbf{k}, n, n) = \frac{Z^2\beta}{4\pi^3\Gamma} \left[ (a^2 + \Gamma^2)\Re\psi_1(z) + \frac{\beta\Gamma}{2\pi}(\Gamma^2 - a^2)\Re\psi_2(z) - \frac{a\beta\Gamma^2}{\pi}\Im\psi_2(z) \right]. \tag{144}$$

Please note that the last two derivations are for illustrative purposes only, as the generic limit (arbitrary $\Phi$) is necessary to check the general expression. For the inter-band magnetic

kernels, the polygamma function has to be expanded to the third order:

$$
\begin{aligned}
\psi_1\left(\frac{1}{2} + \frac{\beta}{2\pi}\left(\Gamma_2 + ia_2\right)\right) = {} & \psi_1\left(\frac{1}{2} + \frac{\beta}{2\pi}\left(\Gamma_1 + ia_1\right)\right) \\
& + \psi_2\left(\frac{1}{2} + \frac{\beta}{2\pi}\left(\Gamma_1 + ia_1\right)\right)\frac{\beta R}{2\pi}\left(\sin(\Phi) + i\cos(\Phi)\right) + \\
& + \psi_3\left(\frac{1}{2} + \frac{\beta}{2\pi}\left(\Gamma_1 + ia_1\right)\right)\frac{\beta^2 R^2}{8\pi^2}\left(\sin(\Phi) + i\cos(\Phi)\right)^2 + \mathcal{O}(R^3)
\end{aligned}
\tag{145}
$$

Due to the massive increase of terms that have to be considered, we confirmed all the Kubo and Boltzmann limits via `Mathematica`.

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
