# Peer review of "LinReTraCe: The Linear Response Transport Centre"

_SciPost Physics Codebases_

## Round 1 · Referee Report · Anonymous (Referee 1) · 2022-7-6

Report

The work by M. Pickem, E. Maggio, and J. M. Tomczak presents a new code to solve the Kubo formula in a semi-analytical way. The manuscript is well written and the code needed in the community as it connects the low temperature quantum incoherence scattering to the high temperature resistive one. It certainly deserves publication.

However there are quite some points that I think the authors should address first: * The authors discussed and compare their approach to the BTE in the relaxation time approximation. However nowadays the de-facto calculations mode is the iterative BTE solution in software such as EPW, Perturbo, Elphbolt or Abinit (Refs. 13,14,17,19 - you might be missing PRB 102, 094308 (2020) for the Abinit). Some of the statements made by the authors do not hold with the iterative solution and need to be discussed. * The LinReTraCe software rely on a linearlization of the electron self-energy. However such linerization (e.g. Eq. 22) is know to be numerically unstable for all but the band edges (e.g. interband transition with low lying energy minima with similar energy). A Dyson-Migdal solution might be more appropriate. How do the authors addressed this instability ? * At the end of Sec. 2, the authors mentioned that the generalization of the band-curvature to the Wannier basis has not yet been derived. What about Eq. 60 in Sec. 3 ? * In Fig. 1, there might be a typo in two blue boxes with "[," * In the footnote 17, page 30, the authors indicate that Boltzmann codes typically use fixed chemical potential or would exhibit "massive" numerical instabilities in gapped system. I disagree. As far as I know, all the BTE codes mentioned above compute the chemical potential using, e.g., a bisection method. It is very stable for the relevant temperature range (e.g. above 50 K). * In section 5.1, the author mentioned that LinReTraCe is more stable that in Boltzmann codes. Could the authors elaborate on that ? In particular, I think it would be very useful to compare the electron/hole conductivity/mobility of a simple semiconductor (e.g. Si) as a function of temperature between LinReTraCe and any of the established BTE codes (they all roughly agree with each other) that computes the transition probabilities/scattering rates. Then accuracy, efficiency and stability can be discussed in details. * The SciPost code acceptance criteria can be found at: https://scipost.org/SciPostPhysCodeb/about#criteria

In particular they specify that benchmarking tests must be provided. I have cloned the code repository at https://github.com/linretrace/linretrace.git As far as I can tell, there are no benchmarking tests, automatic test-suite with reference data etc.
In addition, the userguide.pdf located in linretrace/documentation is very limited (5 pages).

It is also indicated that "At least one example application must be presented in detail". Even if the authors presented some examples in the manuscript, no such example is available with the code and can be run.

I have also looked at the Fortran source files located in linretrace/src_linretrace The main.F90 is relatively well documented (comments in the code) but for other subroutines, it might be improved.

---

## Round 1 · Referee Report · Anonymous (Referee 2) · 2022-8-9

Report

Warnings issued while processing user-supplied markup:

  • Inconsistency: Markdown and reStructuredText syntaxes are mixed. Markdown will be used.
    Add "#coerce:reST" or "#coerce:plain" as the first line of your text to force reStructuredText or no markup.
    You may also contact the helpdesk if the formatting is incorrect and you are unable to edit your text.

Article 202206_00012v1 by Pickem Maggio and Tomczak is an interesting presentation of a new software package to evaluate transport quantities from electronic structure input data. The semi-analytical evaluation of the Kubo integrals comes from a recent PRB, and is implemented in the present software, going beyond standard Boltzmann approaches in allowing for more broadening effects and energy integration of the scattering processes between states.

The code is a new and clean implementation, the presentation is quite thorough, and should be published. There are a number of useful general comments and physical observations about transport which will benefit many practitioners, though they are a bit buried in this very lengthy document.

A few comments about the manuscript and the code:

  • My main complaint would be that the strongest advantages over normal Boltzmann approaches are not really demonstrated or showcased. The lifetime effects and scatterings possible (beyond the energy and T dependent models already in existing codes) do not jump out and shock me. The absence of a low T peak in Tl PbTe is a disappointment, but further remains a mystery: it could be due to yet other extrinsic effects, and the proposed inverse engineering is a bit dangerous in this respect. Any rho(T) curve can be fit with many many scattering models if one allows arbitrary tau(n,k) variations...

  • The solutions of the quasiparticle equations are not always easy to linearize, as the authors require for their technique, especially if correlations are strong. At the end of 1.1 some discussion of the conditions of physical validity of this approach is needed. A "few kBT" can be huge in strongly correlated very narrow d or f bands.

  • p.13: "Generalizations of the band- curvatures to the Wannier basis have not yet been derived" I do not fully understand, though it sounds interesting. BoltzWann implements band curvatures to get the full transport quantities - what is missing or what more do the authors have in mind?

  • Figure 3: I have never had such massive problems in converging chemical potentials in insulators even with small effective temperatures. The proposed refinement is certainly interesting, but I suspect there is some instability which can be fixed with the usual algorithms.

  • I have downloaded and installed the package. On my macbookpro the compilation fails with gcc 11 (cpu does not implement quadruple precision) - I did not debug for very long, but it would be good to have a more robust interface for the quad precision code. On our old cluster things compiled fine with gcc 10.2

  • I tried a few systems from the github repo with examples - these should be mentioned more clearly in the manuscript. The default config.temp files work but produce occasional errors (see below). A more thorough testing of the code and examples is essential. I agree with the other referee that a small test suite is important to ensure the code is well compiled. The runtime for PbTe with 8 MPI threads was actually much longer than I expected (480s), comparing with BoltzTraP for example, though still bearable. The initial claim "as fast as Boltzmann" seems a bit overblown.

======= Overall the manuscript English is excellent, but there are a few typos and a final read through would be beneficial.

page 11 If the same quasi-particle renormalization (no s) page 13 the derivative in k should be outside a parentheses for ∂k U†(k)H(k)U(k) "ensures" not "assures" absent in Eq 55 page 16 explain color code in figure 1 page 17 "the corresponding interface" page 21 The first column (no s) A generalization (no s) page 25 bottom: the phrase concerns a code snippet and not table 1, there is no continuity in the text. page 32 "identically"

======= M1Max chip compilation error cpsipg.f:11:43:

11 | COMPLEX(kind=selected_real_kind (32)) WPSIPG,W | 1 Error: Kind -1 not supported for type COMPLEX at (1)

-> this means precision not supported by the cpu, but quadruple precision is certainly possible. An alternative coding might guarantee universal compilation

PbTe config.temp error: HDF5-DIAG: Error detected in HDF5 (1.10.5) thread 0: #000: H5T.c line 1754 in H5Tclose(): not a datatype major: Invalid arguments to routine minor: Inappropriate type ??

---

## Editorial Decision

resubmitted